# BAYESIAN SELF-DISTILLATION FOR IMAGE CLASSIFICATION

## ABSTRACT

Supervised training of deep neural networks for classification typically relies on hard targets, which promote overconfidence and often limit calibration, generalization, and robustness. Self-distillation methods aim to mitigate this by leveraging inter-class and sample-specific information present in the model's own predictions, but often remain dependent on hard targets, limiting their effectiveness. With this in mind, we propose Bayesian Self-Distillation (BSD), a principled method for constructing sample-specific target distributions via Bayesian inference using the model's own predictions. Unlike existing approaches, BSD does not rely on hard targets after initialization. BSD consistently yields higher test accuracy (e.g. +1.4% for ResNet-50 on CIFAR-100) and significantly lower Expected Calibration Error (ECE) (-40% ResNet-50, CIFAR-100) than existing architecture-preserving self-distillation methods for a range of deep architectures and datasets. Additional benefits include improved robustness against data corruptions, perturbations, and label noise. When combined with a contrastive loss, BSD achieves state-of-the-art robustness under label noise for single-stage, single-network methods. Code is available in the supplementary material.

## 1 INTRODUCTION

Despite the widespread use of deep neural networks for classification tasks, the implications of using hard targets for loss computation have received relatively little attention. Intuitively, classes can exhibit varying degrees of similarity, and individual samples may exhibit different levels of resemblance to both their assigned and other classes. This nuanced, sample-specific information – often referred to as dark knowledge (Hinton, 2014) – is not taken into account when relying solely on hard targets. For instance, misclassifying an image of a dog as a teapot is fundamentally different from misclassifying it as a cat. Nevertheless, deep neural networks are typically trained with hard targets, which fail to capture inter-class relationships or sample-specific ambiguities, contributing to poor calibration (Guo et al., 2017).

*Self-distillation*, an efficient variant of *knowledge distillation* (Hinton et al., 2015), leverages dark knowledge by using a model's own predictions as soft targets during training. In contrast, conventional knowledge distillation provides soft targets to a model by leveraging another, often larger, network. Existing self-distillation methods, however, exhibit varying limitations. Some modify the network architecture with intermediate classifiers (Zhang et al., 2019; 2021), increasing parameter count and computational cost. Others impose a consistency loss between current predictions and those from the last epoch or mini-batch (Kim et al., 2021; Shen et al., 2022). Yet, epoch-wise construction introduces variance under augmentations, while targets from the last mini-batch may be of limited effectiveness, as they originate from nearly identical model states.

Fundamentally, existing self-distillation frameworks constrain their predictions by incorporating a loss term derived from the hard targets (e.g. Furlanello et al. (2018); Kim et al. (2021); Shen et al. (2022)). This reliance interferes with the goal of learning richer and more calibrated predictions, and leaves the model sensitive to label noise. In the overparameterized regime, sensititivity to label noise can exacerbate double descent (Nakkiran et al., 2021), a phenomenon where the test error initially decreases, then increases, before decreasing again as model capacity or training time increase. We hypothesize that a self-distillation method decoupled from the hard targets during training could pro-

Figure 1: **Bayesian update of a target.** At each epoch, the target distributions are updated using the model's own predictions. Here, $\oplus$ denotes the Bayesian update, and $\times$ indicates discounting of the previous posterior parameters by $\gamma \in [0, 1]$. See Section 3 for details.

vide smoother and more robust probability estimates, thereby mitigating these effects and providing more predictable training dynamics.

To address these limitations, we propose Bayesian Self-Distillation (BSD), an efficient method for leveraging dark knowledge by constructing rich, sample-specific target distributions in a single training run. BSD treats a model's own predictions as evidence for Bayesian inference, without explicitly relying on the hard targets after initialization. A diagram depicting BSD is included in Figure 1.

Our main contributions can be summarized as follows:

- We propose Bayesian Self-Distillation (BSD), a lightweight and principled method for self-distillation that operates independently of hard targets after initialization.
- We provide the first, to our knowledge, theoretical formalization of dark knowledge and use it to show that BSD captures rich sample-specific information.
- We demonstrate through extensive experiments that BSD consistently outperforms both existing architecture-preserving self-distillation methods and conventional knowledge distillation in generalization and calibration for a variety of datasets and network architectures.
- We show that BSD provides robustness against data corruptions, perturbations, and label noise. Notably, it mitigates epoch-wise double descent under label noise and achieves state-of-the-art performance for single-stage, single-network methods when combined with a contrastive loss.

## 2 RELATED WORK

Self-Distillation (SD) originates from Knowledge Distillation (KD) (Hinton et al., 2015), a technique introduced for model compression using a teacher–student framework. Later work demonstrated that identical (Born-Again) networks can be trained sequentially with KD to improve generalization (Furlanello et al., 2018), thereby giving rise to self-distillation.

SD methods differ primarily in how they construct the teacher signal. One family of approaches relies on architectural structure, such as introducing auxiliary classifiers or branches to provide internal supervision (Zhang et al., 2019; 2021; Zhu et al., 2018). Another category leverages temporal knowledge transfer, using model snapshots from earlier epochs (Yang et al., 2019), moving averages of past predictions (Temporal Ensembling, TE) (Laine & Aila, 2016), or moving averages of model parameters (mean teacher) (Tarvainen & Valpola, 2017) over epochs. A third set of methods leverages consistency, encouraging alignment between samples of the same class (Yun et al., 2020) or between consecutive predictions of the same sample (Kim et al., 2021; Shen et al., 2022). This category includes Progressive Self-Knowledge Distillation (PS-KD) (Kim et al., 2021), which uses the model's prediction from the last epoch as a soft target, and Self-Distillation from the Last mini-Batch (DLB) (Shen et al., 2022), which constructs overlapping mini-batches and leverages the prediction from the previous mini-batch. Despite their differences, these strategies remain anchored to the original hard labels, which can be particularly problematic in settings where labels are noisy.

Label noise remains an obstacle in deep learning, contributing to overfitting and the undesirable training dynamics. Proposed solutions range from regularization techniques such as label smoothing (Szegedy et al., 2016) and Mixup (Zhang et al., 2017), to robust loss functions like Symmetric

Cross Entropy (Wang et al., 2019). More sophisticated methods include sample selection via optimal transport Feng et al. (2023); Chang et al. (2023), label correction by interpolation between the prediction and label (Xu et al., 2025), or even complex multi-stage (Liu et al., 2023) or multi-network training pipelines (Zhang et al., 2024). These methods, however, often rely on heuristics rather than *reformulating the learning objective to explicitly model uncertainty*. In contrast, probabilistic modeling provides a principled way to estimate uncertainty and improves interpretability.

Bayesian methods in deep learning have gained attention for their ability to model predictive uncertainty. Methods like Variational inference (Blundell et al., 2015), Monte Carlo dropout (Gal & Ghahramani, 2016), and deep ensembles (Lakshminarayanan et al., 2017) can be used to quantify uncertainty, improve calibration and increase robustness to out-of-distribution data. Recently, Bayesian optimization of hyperparameters has been paired with born-again networks to improve generalization over generations of networks. Related work has also shown that KD itself can serve as a mechanism to transfer better calibration from a teacher to a student (Hebbalaguppe, 2024).

Our proposed method, Bayesian Self-Distillation (BSD), draws inspiration from these domains. Like SD, it softens target distributions, but views the problem from a Bayesian perspective where model predictions serve as evidence for updating target distributions. Note that BSD is distinct from both Bayesian neural networks and Bayesian optimization, where the former regards distributions over model weights and the latter is a method for optimizing black-box functions, whereas BSD places priors over target distributions and refines them over epochs. This allows targets to evolve independently without the continual influence of hard targets, providing richer and more flexible self-supervision that inherently offers robustness to label noise.

## 3 METHOD

As discussed in Section 1, a network trained exclusively on hard targets is hardly encouraged to learn meaningful inter-class relationships or account for sample-specific ambiguities. To address this, we frame the training process as Recursive Bayesian Estimation, with the goal of approximating the true latent target distributions $\mathbf{y}$. Rather than viewing the network as a static generator of fixed predictions, we treat it as a noisy sensor observing the external dataset $\mathcal{D}$. By recursively integrating these noisy signals, we aim to construct stable target distributions that capture both class relationships and sample-specific information, thereby improving generalization and reducing overconfidence. The proposed method, Bayesian Self-Distillation (BSD), is summarized in Algorithm 1.

### 3.1 NOTATION

Consider the supervised classification problem with a dataset $\mathcal{D} = \{(\mathbf{x}_i, \mathbf{y}_i^0)\}_{i=1}^n$, samples $\mathbf{x}_i \in \mathbb{R}^d$ and one-hot targets $\mathbf{y}_i^0 \in \Delta_k$. Let $f : \mathbb{R}^d \to \mathbb{R}^k$ denote a neural network with parameters $\boldsymbol{\theta}$ with a softmax activation as its last layer, $\mathcal{L}$ the loss function, and let $\hat{\mathbf{y}}_i^t$ denote the prediction the model outputs for sample $i$ at epoch $t$.

### 3.2 BAYESIAN SELF-DISTILLATION

Training a deep neural network is a stochastic process where randomness enters through stochastic optimization (e.g., SGD), data augmentation, and regularization mechanisms. SGD dynamics approximate a posterior distribution over model parameters (Mandt et al., 2017), while stochastic regularization (e.g. dropout) allows the network's output to be viewed as a draw from a predictive distribution (Gal & Ghahramani, 2016). Consequently, the model's prediction for a sample $\mathbf{x}_i$ at any training step can be seen as a random sample from an implicit predictive distribution.

From this perspective, we treat the network as a noisy sensor measuring the true latent class distribution $\mathbf{y}_i$ of the external input $\mathbf{x}_i$. This interpretation is similar to established frameworks in that it views neural network training as Recursive Bayesian Estimation (Singhal & Wu, 1988). At epoch $t$, the network provides a prediction $\hat{\mathbf{y}}_i^t$, serving as a noisy measurement of the true state $\mathbf{y}_i$. Although the sensor evolves during training making the measurement noise non-stationary, this dynamic is consistent with adaptive filtering and probabilistic self-training frameworks (e.g., Expectation-Maximization). Importantly, this framing allows for model predictions to be viewed as external and independent evidence. The predictions $\hat{\mathbf{y}}_i^t$ constitute measurements as read by the

sensor $f$, conditioned on the independent input $\mathbf{x}_i$. To formalize this, we introduce a latent class variable $z_i \in \{1, \ldots, k\}$ for each sample $\mathbf{x}_i$ and model its distribution as categorical,

$$z_i \sim \text{Cat}(\mathbf{y}_i). \tag{1}$$

where $\mathbf{y}_i = [y_{i,1}, y_{i,2}, \ldots, y_{i,k}]$ represents the probabilities for each of the $k$ classes.

To express prior beliefs about $\mathbf{y}_i$, we use a Dirichlet distribution $\mathbf{y}_i \sim \text{Dir}(\boldsymbol{\alpha}_i)$ due to its conjugacy and representation of accumulated evidence as independent components. Specifically, a Dirichlet distribution can be generated by normalizing a set of independent Gamma variables $v_{i,j} \sim \text{Gamma}(\alpha_{i,j}, 1)$ such that $y_{i,j} = v_{i,j} / \sum_k v_{i,k}$. This allows us to interpret the prior as maintaining independent evidence counters for each class. When the input $\mathbf{x}_i$ triggers a non-zero prediction for a secondary class (e.g., "cat" features in a "dog" image), it acts as an independent sensor update for that specific class. Assuming that the model learns to extract relevant feature representations during training, the predictions will capture semantically meaningful knowledge rather than model hallucinations. We encode prior belief in the class corresponding to the label by letting

$$\alpha_{i,j}^0 = \begin{cases} c, & \text{if } j = \underset{l}{\text{argmax }} y_{i,l}^0, \\ \epsilon, & \text{otherwise,} \end{cases} \tag{2}$$

where $\epsilon \ll c$. For sufficiently small $\epsilon$, the prior implies $p(z_i | \boldsymbol{\alpha}_i^0)_j \approx 1$ for the labeled class.

At each epoch $t$, the model outputs a prediction $\hat{\mathbf{y}}_i^t$ which we treat as a noisy measurement to update our beliefs. We assume the likelihood

$$p\left(\hat{\mathbf{y}}_i | \mathbf{y}_i\right) \propto \prod_{j=1}^{k} y_{i,j}^{\hat{y}_{i,j}}, \tag{3}$$

which generalizes the categorical likelihood to fractional evidence and is conjugate to the prior (Bishop & Nasrabadi, 2006). This likelihood treats the prediction as partial evidence, where higher predicted probabilities correspond to stronger observations.

Formally, after observing a prediction $\hat{\mathbf{y}}_i^t$ at epoch $t$, the posterior distribution is

$$\mathbf{y}_i | \hat{\mathbf{y}}_i^t, \boldsymbol{\alpha}_i^{t-1} \sim \text{Dir}\left(\boldsymbol{\alpha}_i^t\right), \quad \text{where} \quad \boldsymbol{\alpha}_i^t = \boldsymbol{\alpha}_i^{t-1} + \hat{\mathbf{y}}_i^t, \tag{4}$$

which accumulates the noisy measurements into sample-specific distributions over the class probabilities $\mathbf{y}$. To predict the distribution over the labels, we use the posterior predictive distribution. However, because the quality of measurements is expected to improve during training (as the model learns), the measurement noise is non-stationary. Standard Bayesian updating would weight early, noisy observations equally to later, more accurate ones. To address this, we adopt a discounted Bayesian model (West & Harrison, 2006), effectively forgetting old evidence to adapt to the improving network. For a discounting factor $\gamma \in [0, 1]$, we have $\boldsymbol{\alpha}_i^t = \gamma \boldsymbol{\alpha}_i^{t-1} + \hat{\mathbf{y}}_i^t$, yielding the update rules

$$\mathbf{y}_i^t = \frac{\gamma A_i^{t-1}}{\gamma A_i^{t-1} + 1} \mathbf{y}_i^{t-1} + \left(1 - \frac{\gamma A_i^{t-1}}{\gamma A_i^{t-1} + 1}\right) \hat{\mathbf{y}}_i^t, \quad A_i^t = \gamma A_i^{t-1} + 1, \tag{5}$$

for $A_i^t = \sum_{j=1}^k \alpha_{ij}^t$.

Intuitively, the Dirichlet parameters accumulate the model's belief about how much each class is supported by the input $\mathbf{x}_i$. Discounting ensures that more recent predictions are weighted more heavily than earlier, likely worse, ones. Early in training, when predictions are more likely to have high variance, the prior dominates. As training progresses, predictions are expected to become more consistent, providing stronger and more reliable evidence.

### 3.3 RELATIONSHIP BETWEEN BSD AND OTHER METHODS

If initialized at its fixed point $A_i^0 = \frac{1}{1-\gamma}$, the recurrence $A_i^t = A_i^0 = \frac{1}{1-\gamma}$ gives the EMA update

$$\mathbf{y}_i^t = \gamma \mathbf{y}_i^{t-1} + (1 - \gamma) \hat{\mathbf{y}}_i^t. \tag{6}$$

---

**Algorithm 1** Bayesian Self-Distillation (BSD)

---

**Input:** Training set $\mathcal{D} = \{(\mathbf{x}_i, \mathbf{y}_i^0)\}_{i=1}^n$
**Model:** Neural network $f$ with parameters $\boldsymbol{\theta}$, optimizer $h$
**Parameters:** Number of epochs $T$, Dirichlet prior $\boldsymbol{\alpha}_i^0 = [\alpha_{i,1}^0, \ldots, \alpha_{i,k}^0]$, discount factor $\gamma$.

**Initialize:** $A_i^0 = \sum_{j=1}^k \alpha_{ij}^0$
   **for** epoch $t \leftarrow 1$ to $T$ **do**
      **for** mini-batch $B \subseteq \mathcal{D}$ **do**
         $\hat{\mathbf{y}}_i^t \leftarrow f(\mathbf{x}_i; \boldsymbol{\theta}), \quad \forall i \in B$             ▷ Model prediction
         $\mathcal{L} \leftarrow \frac{1}{|B|k} \sum_{i \in B} \ell(\hat{\mathbf{y}}_i^t, \mathbf{y}_i^{t-1})$             ▷ Compute loss
         Compute $\nabla_{\boldsymbol{\theta}} \mathcal{L}$
         $\theta \leftarrow h(\boldsymbol{\theta}, \nabla_{\boldsymbol{\theta}} \mathcal{L})$             ▷ Gradient descent step
         $\mathbf{y}_i^t \leftarrow \frac{\gamma A_i^{t-1}}{\gamma A_i^{t-1}+1} \mathbf{y}_i^{t-1} + \left(1 - \frac{\gamma A_i^{t-1}}{\gamma A_i^{t-1}+1}\right) \hat{\mathbf{y}}_i^t, \quad \forall i \in B$             ▷ Update targets
         $A_i^t = \gamma A_i^{t-1} + 1, \quad \forall i \in B$             ▷ Update prior hyperparameters
      **end for**
   **end for**
**Return** $\theta$             ▷ Trained parameters

---

Similarly,

$$A_i^t = \gamma A_i^{t-1} + 1 = \gamma^t A_i^0 + \sum_{j=0}^{t-1} \gamma^j = A_i^0 \gamma^t + \frac{1-\gamma^t}{1-\gamma} \tag{7}$$

so in the limit we have that

$$\lim_{t \to \infty} A_i^t = \frac{1}{1-\gamma}, \tag{8}$$

i.e. the weight of new observations converges to $1 - \gamma$ exponentially. In the case of non-zero $\epsilon$, and if we initialize at the fixed point $A_i^0 = \frac{1}{1-\gamma}$ for $\gamma$ close to 1, BSD will approximate label smoothing (Szegedy et al., 2016).

Related self-distillation methods can be interpreted as special cases of BSD. Conventional training corresponds to taking $c \to \infty$, which fixes the target distribution and prevents Bayesian updating. In this limit, the framework reduces to standard distribution matching, as KL-divergence and cross-entropy are equivalent under fixed one-hot targets. PS-KD (Kim et al., 2021) is recovered by setting $\gamma = 0$, so that only the most recent prediction contributes to the target. DLB (Shen et al., 2022) performs the same Bayesian update as PS-KD but applies it at the mini-batch level rather than epoch-wise. Meanwhile, TE (Laine & Aila, 2016) corresponds to assuming an improper prior (a zero vector) and applying an explicit weighting schedule to the accumulated evidence. Although these methods differ from BSD in that they include a supervised loss term based on one-hot labels, apply temperature scaling or modify the loss function, all of them implicitly construct a target distribution from an accumulation of past predictions. BSD makes this shared structure explicit by interpreting it as Bayesian evidence aggregation under a Dirichlet prior.

## 4 EXPERIMENTS

**Experimental Setup.** We evaluate ResNet (He et al., 2016a), DenseNet (Huang et al., 2017), and ViT (Dosovitskiy et al., 2020) models on CIFAR-10 (Krizhevsky et al., 2009), CIFAR-100 (Krizhevsky et al., 2009), Tiny ImageNet (Stanford CS231n, 2017) and ImageNet Russakovsky et al. (2015). We report the average of three runs. For BSD, we set $\epsilon = 0$, $\gamma = 0.95$, and use $c = 1000$ and $c = 50$ for the CNNs and ViTs, respectively, on all datasets except for ImageNet. On Imagenet, we set $\epsilon = 0.05$, $\gamma = 0.99$, and use $c = 400$. The experimental setup is described in more detail in the appendix, including the hyperparameters used for the baselines.

## 4.1 EMERGENCE OF DARK KNOWLEDGE

We first validate that dark knowledge emerges naturally during training, and show that BSD promotes further discovery of underlying structures in the data. In an attempt to formalize the notion of dark knowledge, we decompose the output of the network, for an input $\mathbf{x}_i$ with integer label $y_i = \underset{k}{\operatorname{argmax}}\, \mathbf{y}^0_{i,k}$, as

$$f(\mathbf{x}_i) = \boldsymbol{\mu}_{y_i} + \delta(\mathbf{x}_i). \qquad (9)$$

Here, $\boldsymbol{\mu}_{y_i} \in \Delta^K$ is a row of the matrix with elements

$$\mu_{ij} = \frac{1}{2}\left(\mathbb{E}[f(X) \mid Y = i]_j + \mathbb{E}[f(X) \mid Y = j]_i\right) \qquad (10)$$

and captures the inter-class component of dark knowledge under a symmetry constraint. The function $\delta : \mathbb{R}^d \to \mathbb{R}^K$ captures the sample-specific deviation from $\boldsymbol{\mu}_{y_i}$ and is calculated as $\delta(\mathbf{x}_i) = f(\mathbf{x}_i) - \boldsymbol{\mu}_y$.

We visualize $\boldsymbol{\mu}$ and $\delta$ from a ResNet-18 trained on CIFAR-10 in Figures 3 and 4. The log-scale heatmaps of $\boldsymbol{\mu}$ in Figure 3 reveal that the networks learn meaningful inter-class relationships. For instance, they capture similarities between animals but also between bird and airplane, with these clusters being more prominent and the probabilities larger for BSD. The per-sample absolute deviation from $\boldsymbol{\mu}$ presented in Figure 4 validates the emergence of $\delta$ and shows that BSD promotes learning of sample-specific information.

## 4.2 CLASSIFICATION RESULTS

**Methods compared.** We benchmark BSD against conventional training (baseline), Temporal Ensembling (TE) (Laine & Aila, 2016), self-Distillation from Last mini-Batch (DLB) (Shen et al., 2022) and Progressive Self-Knowledge Distillation (PS-KD) (Kim et al., 2021). We do not compare with methods such as (Zhang et al., 2019; 2021), as our focus is on modifying label distributions rather than model architectures. An ensemble of three models is included for reference, whose knowledge is also distilled into a single model of the same architecture, for a comparison with conventional knowledge distillation.

### 4.2.1 GENERALIZATION

The main results are reported in Table 1. In all experiments, BSD improves test accuracy relative to both the baseline and related methods, surpassing conventional knowledge distillation and approaching ensemble performance.

The performance gains are most pronounced on CIFAR-100 and TinyImageNet. For ResNet and DenseNet, BSD boosts accuracy by about 3 percentage points (pp) over the baseline and by more than 1 pp over the strongest related methods on both datasets. In contrast, on CIFAR-10, where baseline performance

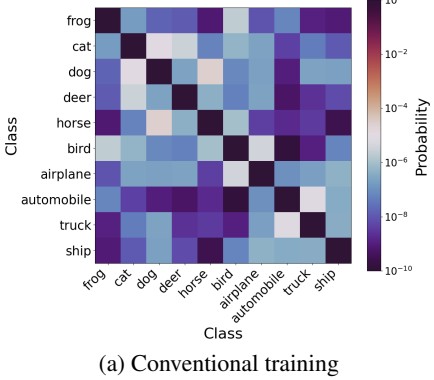

(a) Conventional training

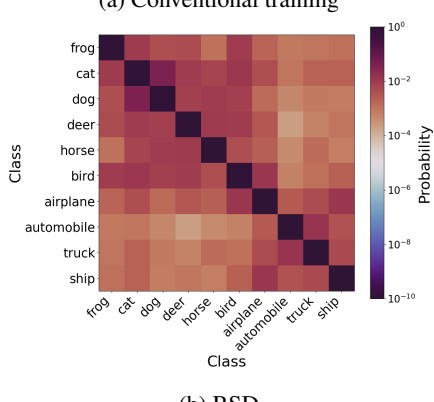

(b) BSD

Figure 3: **Inter-class component $\boldsymbol{\mu}$ of dark knowledge.** Semantical patterns emerge between classes, accentuated by BSD (ResNet-18, CIFAR-10).

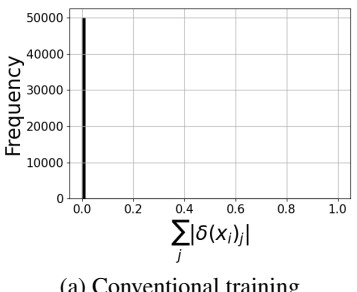

(a) Conventional training

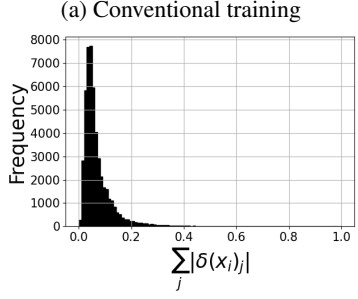

(b) BSD

Figure 4: **Sample-specific component of dark knowledge.** BSD promotes learning of sample-specific information (ResNet-18, CIFAR-10).

Table 1: **Test set accuracy on CIFAR-10, CIFAR-100, and TinyImageNet.** The models are trained using standard training (baseline), related methods (Laine & Aila, 2016; Shen et al., 2022; Kim et al., 2021), and the proposed method (BSD). The best results are highlighted in bold.

| Dataset | Model | Baseline (%) | TE (%) | DLB (%) | PS-KD (%) | BSD (%) | KD (%) | Ens. (%) |
|---|---|---|---|---|---|---|---|---|
| CIFAR-10 | ResNet-18 | $94.39_{\pm0.18}$ | $94.52_{\pm0.08}$ | $94.52_{\pm0.11}$ | $94.56_{\pm0.06}$ | $\mathbf{94.88}_{\pm0.06}$ | $94.73_{\pm0.18}$ | 95.33 |
| | DenseNet-121 | $94.86_{\pm0.13}$ | $95.20_{\pm0.13}$ | $95.15_{\pm0.15}$ | $95.10_{\pm0.04}$ | $\mathbf{95.38}_{\pm0.13}$ | $95.28_{\pm0.09}$ | 95.67 |
| | ViT-B/16 | $98.28_{\pm0.02}$ | $98.34_{\pm0.05}$ | $98.39_{\pm0.03}$ | $98.37_{\pm0.08}$ | $\mathbf{98.44}_{\pm0.01}$ | $98.32_{\pm0.02}$ | 98.56 |
| CIFAR-100 | ResNet-50 | $75.82_{\pm0.41}$ | $76.07_{\pm0.40}$ | $77.22_{\pm0.23}$ | $77.71_{\pm0.20}$ | $\mathbf{79.09}_{\pm0.11}$ | $77.69_{\pm0.08}$ | 79.29 |
| | DenseNet-169 | $76.63_{\pm0.28}$ | $76.33_{\pm0.11}$ | $78.43_{\pm0.09}$ | $77.88_{\pm0.04}$ | $\mathbf{79.47}_{\pm0.12}$ | $78.72_{\pm0.18}$ | 79.83 |
| | ViT-B/16 | $89.16_{\pm0.05}$ | $89.11_{\pm0.23}$ | $88.72_{\pm0.04}$ | $89.36_{\pm0.11}$ | $\mathbf{89.54}_{\pm0.13}$ | $89.35_{\pm0.11}$ | 90.27 |
| TinyImageNet | ResNet-101 | $64.22_{\pm0.21}$ | $64.45_{\pm0.12}$ | $65.83_{\pm0.28}$ | $65.65_{\pm0.12}$ | $\mathbf{67.41}_{\pm0.31}$ | $66.36_{\pm0.16}$ | 69.78 |
| | DenseNet-201 | $64.53_{\pm0.38}$ | $64.60_{\pm0.19}$ | $66.66_{\pm0.20}$ | $66.11_{\pm0.08}$ | $\mathbf{67.74}_{\pm0.04}$ | $66.94_{\pm0.09}$ | 69.44 |
| | ViT-B/16 | $88.99_{\pm0.20}$ | $89.02_{\pm0.23}$ | $89.29_{\pm0.13}$ | $89.32_{\pm0.13}$ | $\mathbf{89.65}_{\pm0.17}$ | $89.16_{\pm0.10}$ | 90.23 |
| ImageNet | ResNet-152 | $78.55_{\pm0.05}$ | $78.45_{\pm0.07}$ | $77.46_{\pm0.14}$ | $78.99_{\pm0.10}$ | $\mathbf{79.47}_{\pm0.07}$ | $79.37_{\pm0.04}$ | 80.24 |

is already high, the improvements are more modest. Similarly, for ViT-B/16, we observe small but consistent gains across all datasets. The trend continues on ImageNet, where BSD improves accuracy by roughly 1 pp over the baseline and 0.5 pp over the best-performing related method, indicating that it also scales effectively to more complex datasets.

### 4.2.2 CALIBRATION

Given the importance of calibrated probability estimates in many tasks, we evaluate the calibration of models trained on CIFAR-100 using Expected Calibration Error (ECE) (Naeini et al., 2015) and Negative Log Likelihood (NLL). The results are included in Table 2, where BSD demonstrates superior calibration when compared to related methods (DLB).

Table 2: **ECE and NLL on CIFAR-100.** The models are trained using standard training (baseline), related methods (Laine & Aila, 2016; Shen et al., 2022; Kim et al., 2021), and the proposed method (BSD). The best results are highlighted in bold.

| Architecture | Metric | Baseline | TE | DLB | PS-KD | BSD |
|---|---|---|---|---|---|---|
| ResNet-50 | ECE (%) | $20.41_{\pm0.45}$ | $20.21_{\pm0.34}$ | $11.82_{\pm0.28}$ | $12.41_{\pm0.17}$ | $\mathbf{7.17}_{\pm0.40}$ |
| | NLL | $2.94_{\pm0.04}$ | $2.91_{\pm0.04}$ | $1.09_{\pm0.01}$ | $1.09_{\pm0.02}$ | $\mathbf{0.77}_{\pm0.00}$ |
| DenseNet-169 | ECE (%) | $19.33_{\pm0.24}$ | $19.63_{\pm0.16}$ | $12.42_{\pm0.09}$ | $12.35_{\pm0.10}$ | $\mathbf{7.67}_{\pm0.10}$ |
| | NLL | $2.59_{\pm0.03}$ | $2.63_{\pm0.01}$ | $1.06_{\pm0.01}$ | $1.08_{\pm0.01}$ | $\mathbf{0.75}_{\pm0.00}$ |
| ViT-B/16 | ECE (%) | $7.53_{\pm0.15}$ | $7.38_{\pm0.26}$ | $6.89_{\pm0.08}$ | $5.93_{\pm0.26}$ | $\mathbf{5.89}_{\pm0.09}$ |
| | NLL | $0.56_{\pm0.02}$ | $0.55_{\pm0.01}$ | $0.52_{\pm0.00}$ | $0.44_{\pm0.01}$ | $\mathbf{0.42}_{\pm0.00}$ |

Figure 5 includes reliability diagrams where BSD's curves lie closest to the diagonal line, indicating better calibration. Furthermore, we compare BSD with standard and distillation-based calibration methods in Table 7 (appendix), where it achieves state-of-the-art results across all metrics.

The improvements are significant compared to the baseline, with BSD reducing ECE by more than 60% and NLL by over 70% for the convolutional networks. Relative to the best-performing related method, BSD lowers ECE by nearly 40% and NLL by about 29%. For ViT-B/16, the gains are more modest, with a 22% reduction in ECE and a 25% reduction in NLL over the baseline, alongside small but consistent improvements over related methods. As discussed in Section 4.2.5, careful hyperparameter selection can further improve BSD's calibration, e.g. an ECE of 1.33% for ResNet-50 on CIFAR-100.

### 4.2.3 ROBUSTNESS

To assess our method's performance under less ideal conditions, we evaluate the models trained on CIFAR-10 on corrupted and perturbed images, and introduce symmetric and asymmetric label noise.

**Corruptions and perturbations.** We evaluate the robustness to corruptions and perturbations of BSD and related methods by evaluating the models trained on CIFAR-10 on CIFAR-10-C and CIFAR-10-P (Hendrycks & Dietterich, 2019). The results are included in Table 3, where we report test set accuracy and mean Flip Probability (mFP) for CIFAR-10-C and CIFAR-10-P, respectively. BSD yields the highest accuracy and the lowest mFP for all models.

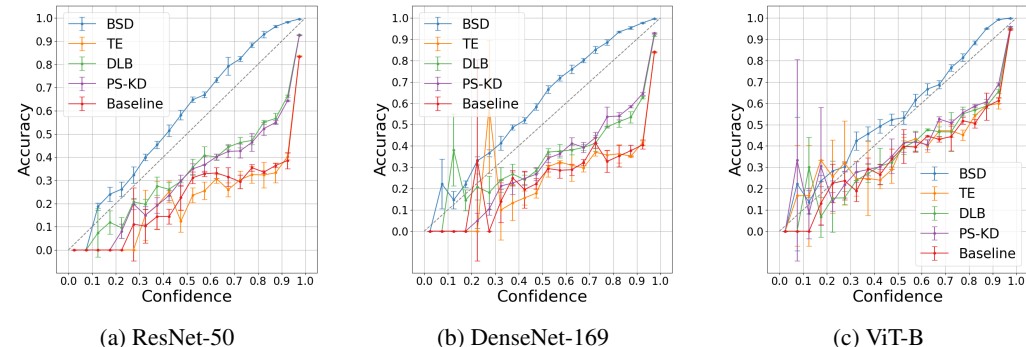

(a) ResNet-50        (b) DenseNet-169        (c) ViT-B

Figure 5: **Reliability Diagrams for CIFAR-100 Models.** A curve closer to the diagonal indicates better calibration. The models are trained using standard training (baseline), related methods (Laine & Aila, 2016; Shen et al., 2022; Kim et al., 2021), and the proposed method (BSD).

Table 3: **Performance on CIFAR-10-C and CIFAR-10-P.** Accuracy and Mean Flip Probability of BSD, baseline, and related methods (Laine & Aila, 2016; Shen et al., 2022; Kim et al., 2021) under corruptions and perturbations. The best results for each metric are highlighted in bold.

| Model | CIFAR-10-C (Acc., %) | | | | | CIFAR-10-P (mFP, %) | | | | |
|---|---|---|---|---|---|---|---|---|---|---|
| | Baseline | TE | DLB | PS-KD | BSD | Baseline | TE | DLB | PS-KD | BSD |
| ResNet-18 | $73.31_{\pm 0.31}$ | $73.22_{\pm 0.27}$ | $73.94_{\pm 0.31}$ | $74.28_{\pm 0.50}$ | $\mathbf{74.88}_{\pm 0.28}$ | $7.20_{\pm 0.21}$ | $7.23_{\pm 0.04}$ | $7.08_{\pm 0.06}$ | $7.10_{\pm 0.28}$ | $\mathbf{6.09}_{\pm 0.08}$ |
| DenseNet-121 | $73.36_{\pm 0.32}$ | $73.09_{\pm 0.86}$ | $74.81_{\pm 0.23}$ | $74.30_{\pm 0.07}$ | $\mathbf{74.85}_{\pm 0.32}$ | $7.49_{\pm 0.20}$ | $7.46_{\pm 0.18}$ | $7.06_{\pm 0.22}$ | $7.21_{\pm 0.16}$ | $\mathbf{6.39}_{\pm 0.11}$ |
| ViT-B | $91.17_{\pm 0.07}$ | $91.31_{\pm 0.22}$ | $91.20_{\pm 0.26}$ | $91.38_{\pm 0.24}$ | $\mathbf{91.57}_{\pm 0.12}$ | $2.53_{\pm 0.04}$ | $2.40_{\pm 0.03}$ | $2.45_{\pm 0.05}$ | $2.35_{\pm 0.06}$ | $\mathbf{2.11}_{\pm 0.07}$ |

On CIFAR-10-C, BSD improves robustness across all architectures, raising accuracy by about $1.5$ pp for both ResNet-18 and DenseNet-121 and by $0.4$ pp for ViT-B over the baseline, slightly surpassing the best related methods. On CIFAR-10-P, BSD achieves the lowest mean flip probability for all models, reducing mFP by around $10\%$ compared to the best-performing related methods.

**Label noise.** We generate symmetric label noise by randomly reassigning a percentage of all labels, and asymmetric noise using the class-dependent definition from prior work (Tanaka et al., 2018). Figure 6 shows test error over epochs for the previously mentioned self-distillation methods under 20% label noise for ResNet-18 trained on CIFAR-10. BSD and TE are obtain the highest accuracy under label noise, while the methods that rely more on the hard targets perform worse. Notably, BSD flattens the characteristic double descent curve observed for other methods, indicating a more robust learning process.

We compare BSD with lightweight regularization methods under noisy labels, including Label Smoothing (LS) (Szegedy et al., 2016), Symmetric Cross Entropy Learning (SL) (Wang et al., 2019), MixUp (Zhang et al., 2017) and Temporal Ensembling (TE) Laine & Aila (2016). We report the average best obtained accuracy, which for BSD is typically close to the final accuracy, but diverges in varying amounts for the remaining methods. The results are included in upper section of Table 4, where BSD yields the highest accuracy for all noise levels except under $10\%$ asymmetric noise. We hypothesize that the strong performance of BSD is due to the complete detachment from the original targets during training, which would explain the robustness under high noise levels.

To compare BSD with state-of-the-art methods for learning with noisy labels, we combine it with techniques from self-supervised learning (for more information refer to the Appendix). We restrict our analysis to single-stage, single-network approaches, as multi-stage or ensemble methods may be used to further enhance BSD's performance. We include OT-Filter (Feng et al., 2023), Curriculum and Structure-aware Optimal Transport (CSOT) (Chang et al., 2023) and Dynamic and Uniform Label Correction (DULC) (Xu et al., 2025). Like other works, we consider a PreAct ResNet-18 (He et al., 2016b), trained with SGD for 300 epochs with a batch size of 128 and weight decay of $5e^{-4}$, learning rate of 0.02 and report the average best obtained accuracy. Results are summarized in the lower section of Table 4, where BSD consistently matches or outperforms existing state-of-the-art methods.

Table 4: **Test set accuracy under symmetric and asymmetric label noise.** Performance of lightweight methods (top, ResNet-18) (Szegedy et al., 2016; Wang et al., 2019; Zhang et al., 2017; Laine & Aila, 2016) and state-of-the-art methods (bottom, Pre-Act ResNet-18) (Feng et al., 2023; Chang et al., 2023; Xu et al., 2025) on CIFAR-10. The best results are highlighted in bold.

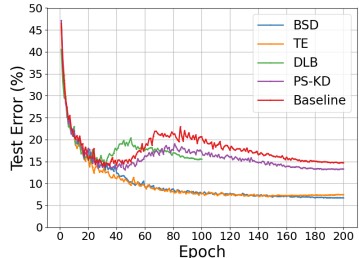

| Method | Symmetric | | | Asymmetric | | |
|---|---|---|---|---|---|---|
| | 20% | 50% | 80% | 10% | 30% | 40% |
| Baseline | $86.76_{\pm0.37}$ | $81.49_{\pm0.26}$ | $63.63_{\pm0.38}$ | $90.07_{\pm0.26}$ | $85.23_{\pm0.29}$ | $80.30_{\pm0.60}$ |
| LS | $87.85_{\pm0.10}$ | $81.49_{\pm0.51}$ | $64.70_{\pm1.17}$ | $90.35_{\pm0.24}$ | $86.10_{\pm0.59}$ | $81.96_{\pm0.52}$ |
| SL | $91.86_{\pm0.12}$ | $86.39_{\pm0.32}$ | $72.89_{\pm0.45}$ | $91.95_{\pm0.14}$ | $86.39_{\pm0.21}$ | $80.57_{\pm0.07}$ |
| MixUp | $89.87_{\pm0.18}$ | $83.38_{\pm0.20}$ | $69.23_{\pm0.56}$ | $91.78_{\pm0.18}$ | $88.26_{\pm0.58}$ | $84.49_{\pm0.54}$ |
| TE | $93.07_{\pm0.07}$ | $90.70_{\pm0.14}$ | $72.61_{\pm0.74}$ | $94.37_{\pm0.15}$ | $92.86_{\pm0.07}$ | $90.81_{\pm0.07}$ |
| BSD | $93.39_{\pm0.05}$ | $90.71_{\pm0.26}$ | $77.50_{\pm0.49}$ | $94.18_{\pm0.07}$ | $93.20_{\pm0.14}$ | $91.65_{\pm0.06}$ |
| OT-Filter | 96.0 | 95.3 | 94.0 | - | - | 95.1 |
| CSOT | $96.6_{\pm0.10}$ | $\mathbf{96.2}_{\pm0.11}$ | $94.4_{\pm0.16}$ | - | - | $95.5_{\pm0.06}$ |
| DULC | 96.6 | 96.0 | 95.0 | 96.7 | 95.5 | 95.2 |
| BSD+ | $\mathbf{96.9}_{\pm0.04}$ | $\mathbf{96.2}_{\pm0.06}$ | $\mathbf{95.1}_{\pm0.06}$ | $\mathbf{97.0}_{\pm0.12}$ | $\mathbf{96.3}_{\pm0.13}$ | $\mathbf{95.6}_{\pm0.25}$ |

Figure 6: **Test Error Over Epochs Under 20% Label Noise on for ResNet-18 on CIFAR-10.** The models are trained using standard training (baseline), related methods (Laine & Aila, 2016; Shen et al., 2022; Kim et al., 2021), and the proposed method (BSD).

### 4.2.4 AUGMENTATION

As data augmentation is a standard component in image classification, we examine the interaction of BSD with two widely used augmentation methods, CutOut (DeVries & Taylor, 2017) and CutMix (Yun et al., 2019). CutOut masks out a patch of an image with zeros, while CutMix replaces the patch with a segment from another image and interpolates the labels. For BSD, the augmentations are applied to half of the images in each mini-batch, while the remaining half is used for distillation. This reduces the frequency of label updates but also changes CutMix to combine soft targets rather than one-hot labels. To compensate, we set $\gamma = 0.9$ and $c = 500$. We include DLB (Shen et al., 2022) and conventional training for comparison. Results are reported in Table 5.

**CutMix.** BSD consistently improves upon both the baseline and DLB across datasets and architectures. On CIFAR-100, BSD boosts convolutional networks by over 3 pp relative to the baseline, and by about $+2.3$ pp (DenseNet-169) and $+1.3$ pp (ResNet-50) compared to DLB. On TinyImageNet, BSD surpasses the baseline by roughly $+2$ pp for ResNet-101 and over $+4$ pp for DenseNet-201, both exceeding DLB by more than 1 pp. For ViT-B/16, BSD achieves gains of $+0.5$ pp over the baseline and $+0.3$ pp over DLB on CIFAR-100, and $+1.15$ pp and $+0.6$ pp on TinyImageNet. On CIFAR-10, improvements are smaller but consistent across all models. Compared with Table 1, CutMix enhances BSD's performance, offering consistent gains over both standard training and DLB.

**CutOut.** The effect of CutOut is more nuanced. On CIFAR-10 and CIFAR-100, BSD consistently performs best, improving convolutional networks by more than $+3$ pp over the baseline and over $+2$ pp over DLB on CIFAR-100. For ViT-B/16, BSD remains competitive with the baseline and performs better than DLB, though the gains are modest given the already high baseline accuracy. On TinyImageNet, however, CutOut does not provide additional benefits: BSD still surpasses the baseline, but the accuracy is lower than without CutOut (cf. Table 1), and ViT-B/16 shows little change relative to baseline or DLB. We speculate that the poor results for BSD on TinyImageNet and DLB overall is due to over-regularization. Overall, BSD remains stronger than DLB under CutOut, but the augmentation itself appears less complementary to BSD than CutMix.

### 4.2.5 ABLATION

We perform ablations on the hyperparameters $\gamma$ and $c$, analyzing their effect on accuracy and ECE for ResNet-50 on CIFAR-100. The results are plotted in Figure 8. We observe high accuracy and low ECE for a range of values. Interestingly, the set of hyperparameters that minimizes ECE does not coincide exactly with the values that maximizes accuracy.

Table 5: **Accuracy on CIFAR-10, CIFAR-100, and TinyImageNet with CutMix and CutOut.** The models are trained using standard training (baseline), DLB (Shen et al., 2022) and the proposed method (BSD). The best results are highlighted in bold.

| Dataset | Model | + CutOut (%) | | | + CutMix (%) | | |
|---|---|---|---|---|---|---|---|
| | | Baseline | DLB | BSD | Baseline | DLB | BSD |
| CIFAR-10 | ResNet-18 | $95.71_{\pm0.09}$ | $95.47_{\pm0.07}$ | $\mathbf{95.94}_{\pm0.07}$ | $95.75_{\pm0.02}$ | $95.70_{\pm0.04}$ | $\mathbf{96.23}_{\pm0.06}$ |
| | DenseNet-121 | $96.09_{\pm0.08}$ | $95.87_{\pm0.12}$ | $\mathbf{96.21}_{\pm0.12}$ | $96.17_{\pm0.20}$ | $96.34_{\pm0.10}$ | $\mathbf{96.84}_{\pm0.08}$ |
| | ViT-B/16 | $98.65_{\pm0.04}$ | $98.58_{\pm0.06}$ | $\mathbf{98.68}_{\pm0.08}$ | $98.82_{\pm0.02}$ | $98.69_{\pm0.04}$ | $\mathbf{98.86}_{\pm0.04}$ |
| CIFAR-100 | ResNet-50 | $76.61_{\pm0.43}$ | $76.38_{\pm0.33}$ | $\mathbf{79.70}_{\pm0.16}$ | $79.73_{\pm0.13}$ | $80.15_{\pm0.30}$ | $\mathbf{81.42}_{\pm0.08}$ |
| | DenseNet-169 | $77.10_{\pm0.17}$ | $77.87_{\pm0.16}$ | $\mathbf{80.14}_{\pm0.35}$ | $79.31_{\pm0.27}$ | $79.93_{\pm0.11}$ | $\mathbf{82.18}_{\pm0.14}$ |
| | ViT-B/16 | $90.30_{\pm0.06}$ | $89.91_{\pm0.10}$ | $\mathbf{90.31}_{\pm0.13}$ | $90.31_{\pm0.05}$ | $90.50_{\pm0.11}$ | $\mathbf{90.87}_{\pm0.04}$ |
| TinyImageNet | ResNet-101 | $64.08_{\pm0.35}$ | $64.26_{\pm0.44}$ | $\mathbf{66.45}_{\pm0.18}$ | $68.32_{\pm0.16}$ | $68.74_{\pm0.14}$ | $\mathbf{70.29}_{\pm0.14}$ |
| | DenseNet-201 | $65.58_{\pm0.17}$ | $64.84_{\pm0.34}$ | $\mathbf{67.62}_{\pm0.21}$ | $66.54_{\pm0.60}$ | $69.22_{\pm0.48}$ | $\mathbf{70.63}_{\pm0.30}$ |
| | ViT-B/16 | $89.63_{\pm0.04}$ | $89.72_{\pm0.03}$ | $\mathbf{89.96}_{\pm0.20}$ | $89.48_{\pm0.03}$ | $90.03_{\pm0.05}$ | $\mathbf{90.63}_{\pm0.14}$ |

The lowest ECE is achieved for $\gamma = 0.97$ and $c = 2000$, while the highest accuracy is observed at $\gamma = 0.96$ and $c = 1000$. Notably, the lowest ECE obtained is $1.33\%$ on the validation set, which is substantially lower than for the other methods in Table 2. While this comes at the cost of a slight decrease in accuracy, calibration may be prioritized for certain tasks or applications. Furthermore, by comparing $\gamma = 1.0$ with the remainder of the values in Figure 8, we note that the discounting is a valuable addition that appears to improve both accuracy and ECE.

While BSD appears offer low ECE and high accuracy for wide range of values of $\gamma$ and $c$, we note that if hyper-parameters are chosen poorly (e.g. $\gamma = 0.9$, $c = 50.0$), performance deteriorates. This is likely due to underfitting caused by relying too much on predictions too early.

(a) Accuracy

(b) ECE

Figure 8: **Impact of $\gamma$ and $c$.** Validation accuracy and ECE for different values of $\gamma$ and $c$ (ResNet-50, CIFAR-100).

## 5 DISCUSSION

Overall, our results demonstrate that BSD can improve test set accuracy, ECE and NLL across a variety of datasets, architectures, and augmentation strategies compared to conventional training and related self-distillation methods. BSD does not seem to overfit to noise in the same way as conventional training and contrastive self-distillation methods, and achieves higher test set accuracy under label noise than other architecture-preserving self-distillation methods. Additionally, BSD+ yields state-of-the-art accuracy under label noise on CIFAR-10.

**Computational cost and memory requirements.** The Bayesian update in Equation 5 requires first computing a per-example scalar weight from $A$ and then applying that weight to the two target tensors. For a mini-batch of size $B$ and $K$ classes, this is $\mathcal{O}(BK)$ and negligible compared with a forward-backward pass. Memory-wise, we store the target distributions $\mathbf{y}$ and per-example counts $A$. With float16 this requires $2NK$ bytes for $\mathbf{y}$ and $2N$ bytes for $A$. TE and PS-KD require the same amount of memory for storing the targets, whereas DLB requires a batch-wise buffer of $2BK$ bytes.

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

# A APPENDIX

## A.1 EXPERIMENTAL DETAILS

Unless otherwise stated, all models trained on CIFAR and TinyImageNet datasets use the Adam optimizer (Kingma, 2014) with a batch size of 256, a maximum learning rate of 0.01 scheduled via the 1cycle policy (Smith & Topin, 2019), and basic augmentations (random cropping and horizontal flipping) for 200 epochs (40 for ViTs, with a maximum learning rate of 5e-5). For ImageNet, we train with SGD for 90 epochs using a per-GPU batch size of 128 on 8 GPUs, a learning rate of 0.5 (scheduled via cosine annealing with 5 epochs of warm-up), a weight decay of $1e-4$, and random resized crops of 224 pixels combined with horizontal flipping.

**Architectures.** Because of the small size of the images, the ResNet and DenseNet networks have been modified to include a $3 \times 3$ convolution instead of the usual $7 \times 7$ convolution for all datasets except ImageNet. The ViT-B (Dosovitskiy et al., 2020) model has been pretrained on ImageNet for all experiments and is available in Pytorch (Paszke et al., 2019).

**Method-specific hyperparameters.** For the temporal ensemble, we set the momentum parameter $\alpha = 0.6$, gradually ramping up the distillation loss over the first 100 epochs (45 for ImageNet) and anneal Adam's $\beta_1$ to zero during the final 50 epochs. For DLB, we follow Shen et al. (2022) and use a temperature of 3 and set $\alpha = 1.0$, but train for only 100 epochs to offset the doubled mini-batch size. For PS-KD, we let $\alpha_T = 0.8$ ($\alpha_T = 0.3$ for ImageNet). For the knowledge distillation of the ensembles, we use a temperature of 3.

**Label Noise.** Inspired by methods in semi-supervised learning, we construct BSD+ by combining BSD with a contrastive loss term. We do this by utilizing a strong and a weak set of augmentations, where the weak set is used for BSD, and the strong is used for the contrastive term. We define the contrastive loss as

$$\mathcal{L}_c = \frac{\lambda_a}{m} \sum_{j=1}^{m} KL(f(T_j(\mathbf{x}_i)), \hat{\mathbf{y}}^t), \quad (11)$$

where $m$ is the number of strongly augmented views, and $T_j$ is the corresponding transform. For the strong set of augmentations, we utilize AutoAugment (Cubuk et al., 2018), in combination with CutMix (Yun et al., 2019) and Random Erasing (Zhong et al., 2020). For BSD+, we set $m = 2$ for all noise levels and schedule the learning rate using cosine annealing (Loshchilov & Hutter, 2016).

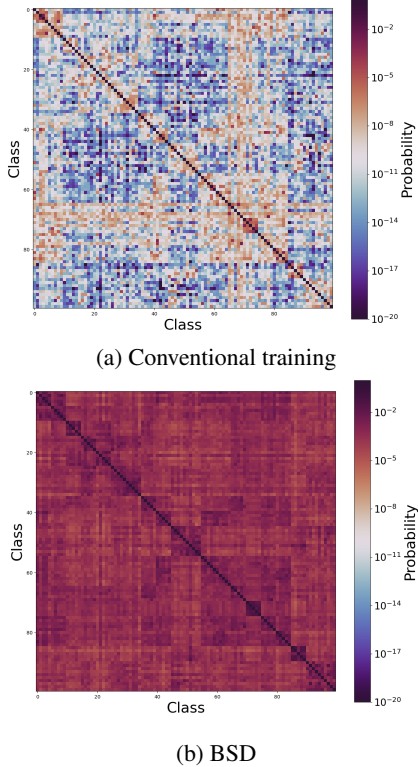

(a) Conventional training

(b) BSD

Figure 10: **Inter-class component $\mu$ of dark knowledge.** Semantical patterns emerge between classes, accentuated by BSD (ResNet-50, CIFAR-100).

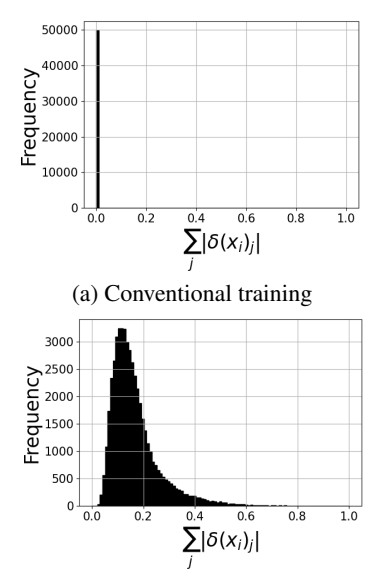

(a) Conventional training

(b) BSD

Figure 11: **Total sample-wise deviation $\delta(\mathbf{x}_i)$ of dark knowledge.** BSD promotes learning of sample-specific information (ResNet-50, CIFAR-100).

Because BSD may excessively smooth the label distributions when labels are noisy, we introduce a sharpening parameter $\tau$ into the loss

$$\mathcal{L} \leftarrow \frac{1}{|B|k} \sum_{i \in B} \ell \left( \hat{\mathbf{y}}_i^t, \frac{\left( \mathbf{y}_i^{t-1} \right)^{1/\tau}}{\| \left( \mathbf{y}_i^{t-1} \right)^{1/\tau} \|_1} \right), \quad (12)$$

for both BSD and BSD+, where $|B|$ is the batch size. In the absence of noise, we use $\tau = 1$ to avoid promoting overconfident predictions, while we set $\tau = 0.8$ for all experiments with label noise. For BSD, we set $\gamma = 0.85$ and $c = 2000$ when injecting symmetric noise, and $\gamma = 0.9$ and $c = 1000$ for asymmetric noise. For BSD+, we set $\gamma = 0.95$ and $c = 1000$ for all experiments, and set $\lambda_c = 2$ under asymmetric noise. For BSD+ under symmetric noise we set $\lambda_c = 4$, $\lambda_c = 7$ and $\lambda_c = 14$ for noise levels 20%, 50% and 80%, respectively.

## A.2 ADDITIONAL EXPERIMENTS

**Emergence of dark knowledge.** We include visualizations of the inter-class distributions for CIFAR-100 and Tiny ImageNet in Figures 10 and 13, with their corresponding sample-wise deviations plotted in Figures 11 and 14, respectively. We observe similar patterns as for ResNet-18 in Figure 3, but on a larger scale, and larger sample-wise deviations for the larger datasets.

To study the emergence of dark knowledge during training, we compute the average KL divergence between the output distributions of the model over epochs and those of the final model, while adjusting for temperature scaling (Guo et al., 2017). The results for are plotted in Figure 15, where the decrease of KL divergence over epochs suggest that dark knowledge is a property that emerges gradually.

**Out-of-distribution calibration and detection.** To evaluate model calibration under distributional shifts, we tested models trained on CIFAR-10 against the CIFAR-10-C benchmark, which applies 19 different corruptions (e.g., brightness, blur, noise) across five severity levels. As shown in Figure 16, BSD consistently achieves the lowest Expected Calibration Error (ECE) across all severity levels compared to the baseline and related methods. Importantly, the performance gap widens as data quality degrades, and we observe that the related methods' ECE increases more rapidly at high corruption severities, while BSD maintains a flatter ECE curve. This indicates that BSD reduces overconfidence under increasing covariate shift.

Furthermore, we study the performance of BSD and related methods under domain shifts by measuring the area under the receiver operating characteristic curve (AUROC) for models trained on CIFAR-10 and CIFAR-100, evaluated against the Street View House Numbers (SVHN) dataset Netzer et al. (2011). While the best performing method varies with dataset and

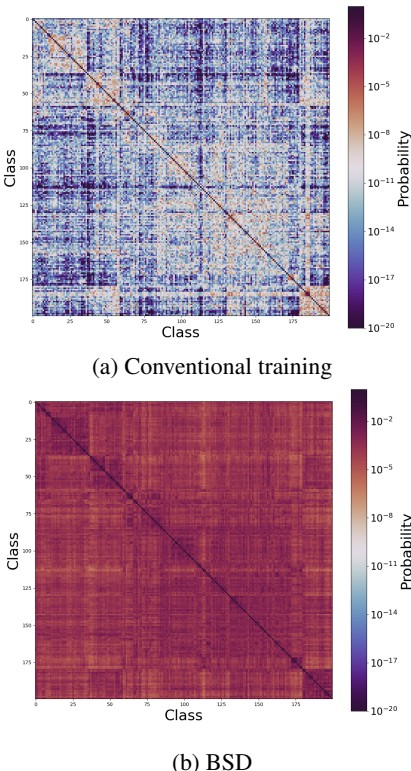

(a) Conventional training

(b) BSD

Figure 13: **Inter-class component $\mu$ of dark knowledge.** Semantical patterns emerge between classes, accentuated by BSD (ResNet-101, Tiny ImageNet).

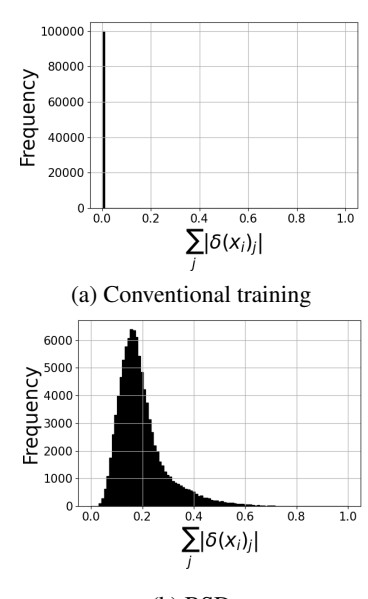

(a) Conventional training

(b) BSD

Figure 14: **Total sample-wise deviation $\delta(\mathbf{x}_i)$ of dark knowledge.** BSD promotes learning of sample-specific information (ResNet-101, Tiny ImageNet).

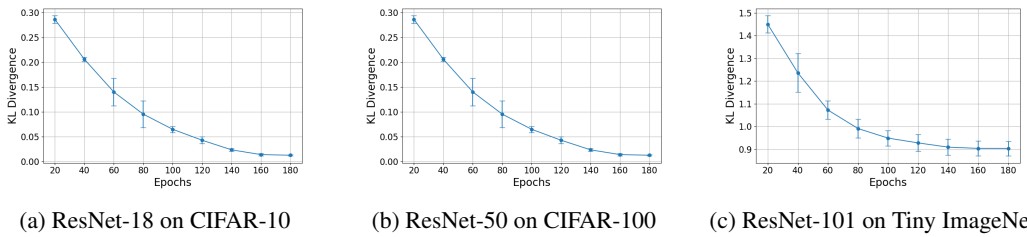

(a) ResNet-18 on CIFAR-10    (b) ResNet-50 on CIFAR-100    (c) ResNet-101 on Tiny ImageNet

Figure 15: **Evolution of the average Temperature-Adjusted KL Divergence of predictions between the current and final model.** Dark knowledge emerges gradually during training.

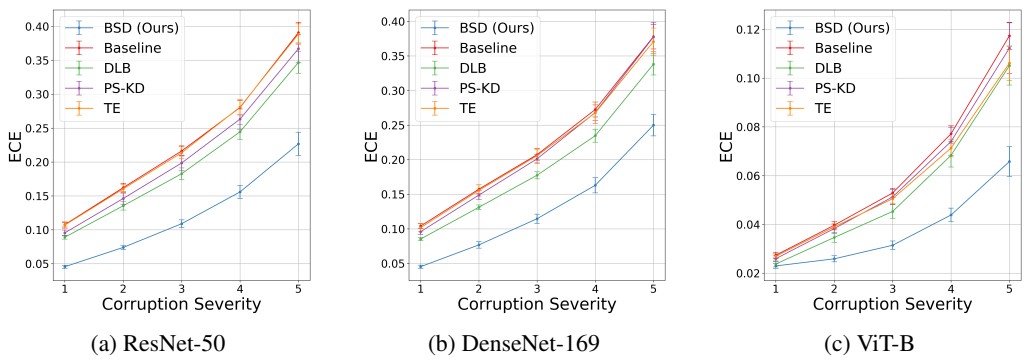

(a) ResNet-50    (b) DenseNet-169    (c) ViT-B

Figure 16: **ECE under increasing corruptions.** The models are trained using standard training (baseline), related methods (Laine & Aila, 2016; Shen et al., 2022; Kim et al., 2021), and the proposed method (BSD), on CIFAR-10 and evaluated on CIFAR-10-C.

Table 6: **Out-of-Distribution Detection Performance (AUROC).** Models trained on in-distribution datasets (CIFAR-10, CIFAR-100) are evaluated against SVHN as the out-of-distribution dataset. The models are trained using standard training (baseline), related methods (Laine & Aila, 2016; Shen et al., 2022; Kim et al., 2021), and the proposed method (BSD). The best results are highlighted in bold.

| ID Dataset | Model | Baseline (%) | TE (%) | DLB (%) | PS-KD (%) | BSD (%) |
|---|---|---|---|---|---|---|
| CIFAR-10 | ResNet-18 | $90.67_{\pm3.42}$ | $92.90_{\pm0.66}$ | $91.25_{\pm0.57}$ | $92.36_{\pm0.48}$ | $\mathbf{94.84}_{\pm0.83}$ |
| | DenseNet-121 | $90.96_{\pm4.22}$ | $\mathbf{94.05}_{\pm0.98}$ | $91.19_{\pm1.20}$ | $91.11_{\pm0.64}$ | $93.65_{\pm0.16}$ |
| | ViT-B/16 | $98.21_{\pm0.29}$ | $\mathbf{98.35}_{\pm0.56}$ | $98.15_{\pm0.22}$ | $98.28_{\pm0.50}$ | $98.03_{\pm0.21}$ |
| CIFAR-100 | ResNet-50 | $74.35_{\pm2.23}$ | $74.68_{\pm1.53}$ | $\mathbf{82.51}_{\pm0.61}$ | $76.66_{\pm2.40}$ | $78.19_{\pm1.19}$ |
| | DenseNet-169 | $77.92_{\pm0.57}$ | $68.97_{\pm6.85}$ | $80.55_{\pm3.56}$ | $73.95_{\pm2.06}$ | $\mathbf{82.41}_{\pm1.79}$ |
| | ViT-B/16 | $88.42_{\pm1.19}$ | $89.12_{\pm0.64}$ | $89.52_{\pm1.16}$ | $88.82_{\pm0.39}$ | $\mathbf{90.71}_{\pm0.50}$ |

model architecture, we observe that BSD yields the most consistent improvement across datasets and architectures. Notably, while TE exhibits instability under this shift on CIFAR-100 (e.g., degrading performance on DenseNet-169 trained on CIFAR-100 with respect to baseline), BSD maintains robust performance. For ViT-B, we observe performance saturation on CIFAR-10 (with all methods $> 98\%$), while for CIFAR-100, BSD yields a notable improvement ($+2.29\%$) over the baseline.

**Calibration.** We train a WideResNet-40-1 on CIFAR-100 to benchmark BSD against different calibration methods including distillation-based method. We compare against Label Smoothing (LS) (Szegedy et al., 2016), Temperature Scaling (TS) (Guo et al., 2017), MixUp (Zhang et al., 2017), Correctness Ranking Loss (CRL) (Moon et al., 2020), PS-KD (Kim et al., 2021), Multi-class Difference in Confidence and Accuracy (MDCA) (Hebbalaguppe et al., 2022), AdaFocal (Ghosh et al., 2022), Calibration by Pairwise Constraints (CPC) (Cheng & Vasconcelos, 2022), Margin-based Label Smoothing (MbLS) (Liu et al., 2022), Adaptive and Conditional Label Smoothing (ACLS) (Park et al., 2023) and combinations of the aforementioned method with knowledge distillation (Hebbalaguppe, 2024). The results in included in Table 7, where we observe that BSD yields the highest

Table 7: **Calibration Performance for WideResNet-40-1 on CIFAR-100.** Result for all methods (excl. BSD and baseline) are from (Hebbalaguppe, 2024). The best results are highlighted in bold.

| Method | Accuracy (%) | ECE (%) | SCE (%) | ACE (%) |
|---|---|---|---|---|
| Baseline (NLL) | 70.04 | 11.16 | 0.30 | 11.19 |
| LS | 70.07 | 1.30 | 0.21 | 1.49 |
| TS | 70.04 | 2.57 | 0.19 | 2.50 |
| MMCE | 69.69 | 7.34 | 0.25 | 7.37 |
| MixUp | 72.04 | 2.57 | 0.21 | 2.52 |
| CRL | 65.80 | 13.91 | 0.37 | 13.91 |
| PS-KD | 72.56 | 3.73 | 0.20 | 3.72 |
| MDCA | 68.51 | 1.35 | 0.21 | 1.34 |
| AdaFocal | 67.36 | 2.10 | 0.21 | 1.97 |
| CPC | 69.99 | 7.61 | 0.23 | 7.55 |
| MBLS | 69.97 | 5.37 | 0.22 | 5.37 |
| ACLS | 69.92 | 7.00 | 0.23 | 6.99 |
| KD | 69.60 | 15.18 | 0.37 | 15.18 |
| KD + MixUp | 72.48 | 1.21 | 0.20 | 1.17 |
| KD + AdaFocal | 71.70 | 1.19 | 0.19 | 1.34 |
| KD + CPC | 70.00 | 9.02 | 0.26 | 9.01 |
| KD + MDCA | 71.07 | 0.98 | 0.20 | 1.10 |
| KD + MMCE | 72.08 | 2.02 | 0.19 | 1.95 |
| BSD ($\gamma = 0.97, c = 2000$) | 72.34$_{\pm 0.26}$ | **0.85** $_{\pm 0.08}$ | **0.18** $_{\pm 0.00}$ | **0.87** $_{\pm 0.19}$ |
| BSD ($\gamma = 0.96, c = 3000$) | 72.62 $_{\pm 0.20}$ | 3.28$_{\pm 0.04}$ | 0.19$_{\pm 0.00}$ | 3.19$_{\pm 0.12}$ |
| BSD ($\gamma = 0.95, c = 4000$) | **72.79** $_{\pm 0.45}$ | 7.08$_{\pm 0.40}$ | 0.23$_{\pm 0.00}$ | 7.08$_{\pm 0.40}$ |

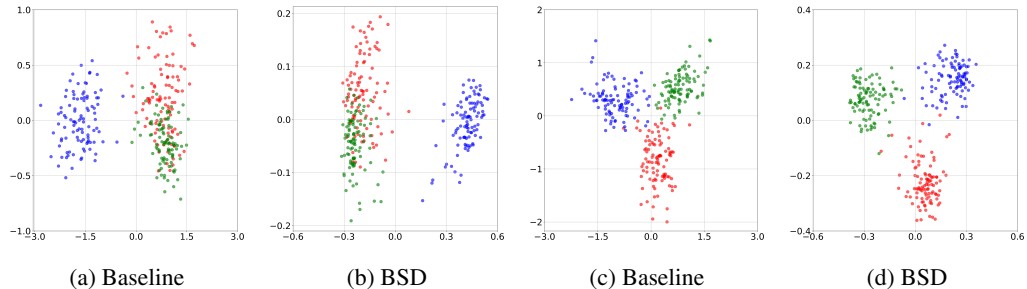

|  (a) Baseline | (b) BSD | (c) Baseline | (d) BSD |

Figure 17: **Penultimate layer representations of ResNet-50 on CIFAR-100.** a-b: two semantically similar classes with one dissimilar class. c-d: Three semantically dissimilar classes.

accuracy and the lowest ECE, Static Calibration Error (SCE) and Adaptive Calibration Error (ACE) of all methods.

### A.3 PENULTIMATE LAYER REPRESENTATIONS

Inspired by Müller et al. (2019), we visualize the penultimate layer representations in Figure 17. BSD yields tighter, less overlapping clusters than conventional training, which is somewhat surprising since BSD promotes learning of sample-specific features. It seems that learning similarities between classes can help differentiate among them.

### A.4 LIMITATIONS

While we experiment with various forms of data augmentation, the interaction with different augmentation schemes as well as regularization techniques warrants further study. Intuitively, augmentations that increase prediction variance may benefit from higher values of the discount factor $\gamma$. Finally, BSD requires selecting the discount factor $\gamma$ and the prior strength $c$, which, despite the observed performance across a large range of settings, could be viewed as a methodological limitation.

## A.5 THE USE OF LARGE LANGUAGE MODELS

We utilized Large Language Models (LLMs) in a limited capacity to improve the quality of the paper. Specifically, LLMs assisted in writing to improve clarity and grammar, suggested related works for us to consider, and were used as a tool for general feedback. The research ideas, experiments, and methodological design were conceived and carried out by the authors. LLMs did not contribute new scientific insights or results.

