# OpenReview forum: "Bayesian Self-Distillation for Image Classification"
_ICLR.cc/2026/Conference — Submitted to ICLR 2026_

### Official Review · Reviewer_cnVX · 2025-10-25

**Soundness:** 3
**Presentation:** 2
**Contribution:** 2
**Rating:** 4
**Confidence:** 4

**Summary:**

This paper proposes Bayesian Self-Distillation (BSD), which integrates ideas from self-distillation and evidence-based (Dirichlet) learning. Instead of using hard labels, BSD updates per-sample target distributions through a Bayesian process that treats model predictions as evidence. The method no longer relies on hard targets after initialization and is evaluated on CIFAR-10/100 and Tiny ImageNet. Experiments show moderate improvements in test accuracy and Expected Calibration Error (ECE), with additional robustness to label noise and data corruption.

**Strengths:**

1. Proposes a principled probabilistic view of self-distillation using Dirichlet updates.
2. The method is lightweight and can be easily integrated into existing pipelines.
3. Provides theoretical framing for “dark knowledge” and sample-specific uncertainty.
4. Well-written and reproducible; includes clear algorithmic steps and ablation on γ and c.

**Weaknesses:**

1. Limited improvement: Accuracy and calibration gains (ECE/NLL) are modest, often within 1–2 pp or comparable to label smoothing or Mixup which both also designed for acc and calibration.
2. Since the objective is to learn the calibrated confidence distribution, maybe training with a calibration focus loss such as [1] may result in a more reliable dirlclet distribution.
2. Small-scale evaluation: Only CIFAR-10/100 and TinyImageNet are used; ImageNet-scale or domain-shift (OOD) results are missing.
3. Calibration not tested on OOD data: A model trained with soft labels should ideally generalize to distributional shifts, but this is not evaluated.
4. Missing comparison to smoothing baselines: Methods such as Label Smoothing, Adaptive LS as BSD appears closely related.
5. No warm-up or convergence analysis: The method may benefit from warm-up epochs or temperature schedules; this is not explored.
6. Unclear practical gain: Given similar computational cost, it is unclear when BSD would be preferred over simpler smoothing strategies.

[1] Dual focal loss for calibration

**Questions:**

As before

---

> ### Author Response · Authors · 2025-11-22
> **We thank Reviewer cnVX for reviewing our work**
>
> We thank the reviewer for their feedback. We have expanded the evaluation in the revised version of the paper to address the concerns regarding scale and OOD performance, and we provide clarification regarding the calibration baselines.
>
> **1. Comparison with Calibration Baselines**
> Regarding the utility of BSD compared to Label Smoothing, MixUp, and other calibration methods, we would like to draw the reviewer’s attention to Table 7 in the Appendix of the revised paper (Table 6 in the original submission).
>
> * **Content:** This table explicitly benchmarks BSD against 10+ other methods, including AdaFocal, Label Smoothing (LS), Margin-based Label Smoothing (MbLS), and MixUp.
> * **Results:** BSD achieves the lowest ECE ($0.85$%), significantly outperforming AdaFocal ($2.10$%) and Label Smoothing ($1.30$%) and the other methods. Specifically, BSD improves ECE by over 13%, SCE by over 5%, and ACE by 20% over the SOTA methods listed.
>
> **2. The Choice of Loss Function**
> We thank the reviewer for the suggestion of using a calibration-focused loss such as dual focal loss (Tao et al., 2023). However, BSD’s goal is not to directly optimize calibration metrics, but to match distributions between the model’s prediction and the aggregated soft target. KL divergence is a natural fit for this purpose, as it measures discrepancy between two full probability distributions. In practice, KL-based matching already yields strong calibration results (Table 7), without requiring additional calibration-specific objectives.
>
> **3. Scale and OOD Evaluation**
> * **ImageNet:** To address concerns regarding scale, we added ImageNet experiments (Table 1). BSD achieves $79.47$% top-1 accuracy with ResNet-152, outperforming both the baseline ($78.55$%) and the best performing related method (PS-KD, $78.99%$). This demonstrates that BSD scales effectively to more complex datasets.
> * **OOD Calibration (Covariate Shift):** To address the lack of OOD testing, we now measure calibration for each severity level of CIFAR-10-C (Figure 16). BSD maintains significantly lower ECE across all corruption severities compared to baselines, indicating superior reliability under distributional shift.
> * **OOD Detection:** We also added Table 6, evaluating OOD detection performance (AUROC) against the SVHN dataset for models trained on CIFAR-10 and CIFAR-100. While the results are mixed across architectures and datasets, we see that BSD is the most consistent method tested.
>
> **4. Warm-up and Convergence**
> Regarding warm-up, we clarify that the prior strength parameter $c$ in combination with the discounting rate $\gamma$ effectively acts as an implicit warm-up mechanism. A higher $c$ enforces stronger adherence to the initialization early in training, while $\gamma$ controls the rate of forgetting. We also wish to refer the reviewer to Section 3.3 (Eq. 11), where our limit calculation shows that the weight of new observations converges exponentially, providing an implicit schedule.
>
> **5. Practical Gain**
> Regarding the comment on practical gain, we would like to point out that BSD offers substantial practical benefits:
> * **Accuracy:** BSD consistently improves accuracy over related methods, surpassing the performance of conventional knowledge distillation.
> * **Calibration:** It achieves large reductions in ECE (e.g., -40% for ResNet-50 on CIFAR-100) compared to the best performing related method.
> * **Robustness:** BSD yields state-of-the-art robustness under label noise for single-stage methods (Table 4).
>
> We thank the reviewer again for their time and insights. We hope that the revised paper and this response have adequately addressed the reviewer's concerns.

---

> > ### Comment · Reviewer_cnVX · 2025-11-25
> >
> > Thank you for the detailed rebuttal and the additional experiments. I understand that the rebuttal period is short and it is difficult to produce a fully comprehensive and polished set of new experiments within this limited time. However, given the scale and nature of my concerns, I believe the paper would benefit from more complete and thoroughly validated results before publication. For this reason, I prefer to keep my original score. I encourage the authors to further strengthen the experimental evaluation and consider resubmitting to a future venue.
> >
> > And training with calibration loss is a good experiment to add to see if this improves.

---

### Official Review · Reviewer_s5Y5 · 2025-11-01

**Soundness:** 3
**Presentation:** 3
**Contribution:** 2
**Rating:** 6
**Confidence:** 3

**Summary:**

This paper introduces a novel training method termed Bayesian Self-Distillation (BSD), which employs a principled Bayesian framework to dynamically construct and evolve sample-specific soft target distributions during training, completely eliminating reliance on conventional hard targets after initialization. The core innovation of BSD lies in treating the model's own predictions as Bayesian evidence, enabling efficient updates via a conjugate Dirichlet prior and incorporating a discount factor to prevent target ossification. This approach allows the model to more effectively uncover and leverage inter-class relationships and sample-specific information called ‘dark knowledge’ inherent in the data, while inherently conferring strong robustness to label noise. Extensive experimental validation demonstrates that BSD consistently enhances the generalization capability and probability calibration of diverse deep architectures across multiple datasets, significantly outperforming existing mainstream self-distillation methods. Furthermore, while maintaining manageable computational cost, BSD achieves state-of-the-art performance among single-stage, single-network methods in handling noisy labels, thereby offering a theoretically grounded and practically effective solution for building more reliable and robust deep learning models.

**Strengths:**

1. This paper demonstrates outstanding performance in theoretical elegance and experimental rigor. Its proposed Bayesian Self-Distillation framework establishes a theoretical system through rigorous mathematical derivation by adopting a Bayesian perspective to treat model predictions as dynamic evidence, leveraging the conjugate relationship between Dirichlet priors and multinomial likelihoods to derive efficient target update formulas, while innovatively introducing a discount factor to resolve confidence accumulation rigidity.

2. This study designs a systematic and comprehensive evaluation scheme, not only compares BSD against traditional training and various self-distillation methods on accuracy and calibration metrics but also analyzes the impact of key hyperparameters through ablation studies, thoroughly investigates synergistic effects with different data augmentation strategies, and visually demonstrates BSD's enhanced capability to mine inter-class structures and sample-specific features. Particularly commendable is the detailed analysis of computational efficiency and memory overhead, coupled with candid discussions of current limitations and future directions.

**Weaknesses:**

1. The authors themselves acknowledge that its effectiveness on very large-scale datasets, such as ImageNet, remains unvalidated, and the introduction of new hyperparameters like prior strength and discount factor complicates model interpretability. In the context of noisy labels, the Temporal Ensembling method proposed in 2016 surprisingly achieves comparable performance to BSD, which contradicts the paper's emphasis on "strong robustness to label noise" as a core advantage and raises doubts about the actual superiority of its noise resistance capability.

2. Furthermore, this self-distillation approach, which relies entirely on the model's own predictions to update targets, is highly susceptible to cognitive rigidity and error amplification. If the model forms erroneous representations early due to data biases or training instability, BSD’s iterative mechanism continuously reinforces these deviations, which is a critical issue that the paper seemingly fails to address. Additionally, the source code for this paper has not been made publicly available, making it difficult to examine the implementation details of the method.

**Questions:**

1.  The paper states in its introduction that methods using either soft or hard labels are significantly affected by noisy labels. Why does Temporal Ensembling (TE) from 2016, which also uses soft labels, achieve strong performance in the noisy label tests?

2.  Self-distillation should, in theory, suffer from error amplification due to incorrect model predictions. Shouldn't this issue be explicitly discussed?

3.  In Figure 6, why was the DLB method only trained for 100 epochs, while all other methods were trained for 200 epochs?

---

> ### Author Response · Authors · 2025-11-22
> **We thank Reviewer s5Y5 for reviewing our work**
>
> We thank the reviewer for the encouraging assessment of the theoretical derivation of our method and experimental rigor. We address their specific questions below.
>
> **1. Validation on Large-Scale Datasets (ImageNet)**
> The reviewer correctly noted that effectiveness on large-scale datasets was unvalidated. We have addressed this by adding ImageNet results in Table 1 of the revised version of the paper.
> * **Results:** BSD achieves $79.47$% top-1 accuracy with ResNet-152, outperforming both the baseline ($78.55$%) and the best performing related method (PS-KD, $78.99$%). This demonstrates that BSD scales effectively to more complex datasets.
>
> **2. Hyperparameters and Interpretability**
> While the introduction of hyperparameters can be seen as a weakness, we demonstrate in the ablation study (Section 4.2.5) that BSD maintains high accuracy and low ECE across a wide range of $\gamma$ and $c$ values (Figure 8), suggesting that it is not sensitive to the choice of hyperparameters.
>
> **3. Robustness and Temporal Ensembling (TE)**
> We would like to point out that while TE is comparable at lower noise levels, Table 4 shows that BSD outperforms TE as the noise level increases (e.g., $77.50$% vs $72.61$% at 80% symmetric noise). We attribute this to BSD's ability to decouple completely from the noisy hard targets after initialization, whereas TE maintains dependence on them.
>
> **4. Cognitive Rigidity and Error Amplification**
> We thank the reviewer for raising a valid point about potential error amplification and for the opportunity to clarify our reasoning. In short, we view this as a hyperparameter dynamic rather than a fundamental flaw. Furthermore, for error amplification to take place, predictions must get exceedingly worse, which is not what we observe.
>
> * **Prior Strength ($c$):** A sufficiently strong prior prevents the model from collapsing into early errors (cognitive rigidity) by keeping targets close to initialization until predictions become reliable.
> * **Ablation Support:** As shown in the ablation study (Figure 8), performance is stable across a wide range of values. Performance only deteriorates if parameters are chosen poorly, leading to underfitting rather than error amplification. We have expanded the discussion in Section 4.2.5 of the revised paper to address this.
>
> **5. Source Code**
> We have added the source code to the supplementary material of the revised paper.
>
> **6. Clarification on DLB Epochs**
> Regarding the question on why DLB was trained for 100 epochs in Figure 6: We followed the original DLB implementation (Shen et al., 2022), which uses overlapping mini-batches (effectively seeing data twice per epoch). Thus, 100 epochs of DLB is computationally equivalent to 200 epochs of standard training.
>
> We thank the reviewer again for their time and insights. We hope that the revised paper and this response have adequately addressed the reviewer's concerns.

---

> > ### Author Response · Authors · 2025-11-27
> > **Follow-up on ImageNet experiments and error amplification**
> >
> > We thank the reviewer once again for their helpful comments. We would like to ask the reviewer to get back to us if their concerns have not been fully addressed. Specifically, we wanted to highlight that we have included the requested ImageNet experiments (Table 1) and addressed the reviewer’s concern regarding error amplification. We remain available to answer any further questions or provide additional clarifications before the discussion period closes.

---

### Official Review · Reviewer_aqga · 2025-11-03

**Soundness:** 1
**Presentation:** 1
**Contribution:** 2
**Rating:** 4
**Confidence:** 2

**Summary:**

This paper introduces Bayesian Self-Distillation (BSD), a novel method for self-distillation in image classification. The core idea is to move away from supervision signals based on one-hot hard targets and instead use soft targets. The method frames the problem through a Bayesian inference lens: it models the latent class distribution with a Dirichlet prior and, at each step, uses the model's own current prediction as evidence to perform a Bayesian update, yielding a posterior distribution. This posterior, modulated by a discount factor to weigh new predictions more heavily over time, then serves as the new supervision signal. The experimental evaluation demonstrates that BSD achieves some improvements in accuracy, calibration (ECE), and robustness to corruptions and label noise across multiple datasets (CIFAR-10/100, Tiny ImageNet) and architectures (ResNet, DenseNet, ViT) when compared to other self-distillation techniques.

**Strengths:**

1.	The core idea of treating the target distribution as a random variable to be inferred via a Bayesian update is novel in the context of self-distillation. Decoupling the training process from the original hard targets after initialization is a principled approach to leveraging the model's "dark knowledge" and allows for the creation of richer, sample-specific supervisory signals.
2.	The paper is supported by comprehensive empirical evidence. BSD consistently outperforms existing architecture-preserving self-distillation methods in terms of raw accuracy, particularly on more complex datasets like CIFAR-100 and TinyImageNet. The improvements in model calibration are large. Furthermore, the method demonstrates superior robustness against data corruptions, perturbations, and especially label noise.

**Weaknesses:**

1.	The central weakness of the paper is the conceptual justification for using the model's own prediction as "observation" or "evidence" in the Bayesian update rule. Bayesian inference is about updating a prior belief based on new evidence from the external world (i.e., observed data). In this work, the "evidence" is generated by the model itself. This is conceptually analogous to forming a belief about a coin's bias, and instead of actually flipping the coin to collect data, one simply imagines the outcome of a flip and uses that imagined outcome to update the belief. This fundamental step requires a much stronger justification than is currently provided.
2.	The presentation of the method is overly reliant on mathematical formalism and fails to build sufficient intuition for the reader. The paper lacks a clear, high-level motivation for why this specific Bayesian framework is the right tool for the job. The method section consists primarily of a sequence of formulas without concrete examples to make the process tangible. The paper would be improved by focusing more on explaining the intuition behind the formulas.

**Questions:**

1.	Regarding Weakness 1: Could you provide a more detailed and intuitively acceptable explanation for why the model's current prediction can be treated as valid, external evidence within a Bayesian framework? I will raise my score if this point can be satisfactorily addressed.

---

> ### Author Response · Authors · 2025-11-22
> **We thank Reviewer aqga for reviewing our work**
>
> We thank the reviewer for their insightful critique, particularly regarding the justification of BSD, which helped us refine our method conceptually.
>
> **1. Justification of the Bayesian Framework (Soundness)**
> In the revised version of the paper (Section 3.2), we clarified the methodology by grounding the mathematical formulation with concrete intuition.
>
> * **Training as a Stochastic Process:** We have expanded the text to explicitly frame training not as convergence toward a fixed point, but as a stochastic process driven by SGD dynamics, random augmentations, and stochastic regularization. Consequently, the model parameters at any specific step represent just one point on a trajectory through the loss landscape.
> We explicitly clarify that the method views the model's prediction at any given step not as static "truth," but as a single draw from the model's implicit predictive distribution.
> * **Related Works:** This interpretation aligns with established work, such as Monte Carlo dropout (Gal & Ghahramani, 2016) and Deep Ensembles (Lakshminarayanan et al., 2017), which treat training/inference stochasticity as sampling.
> * **Motivation:** We have updated Section 3.2 to explicitly motivate why this framework is meaningful in this context: it provides a principled mechanism to aggregate these stochastic predictions into sample-specific target distributions while accounting for prior knowledge.
>
> **2. Intuition vs. Formalism**
> Following the reviewers suggestion, we have clarified the methodology to provide better intuition for the reader. For example, we now explicitly describe the Dirichlet parameters $\alpha$ as accumulated evidence, and the discount factor $\gamma$ as a mechanism to forget early, unreliable predictions, preventing the ossification of targets.
>
> We thank the reviewer again for their time and insights. We hope that the revised paper and this response have adequately addressed the reviewer's concerns.

---

> > ### Comment · Reviewer_aqga · 2025-11-26
> >
> > Thank you for your response.
> >
> > Unfortunately, my primary concern regarding the soundness of the "Bayesian" framework remains unaddressed.
> > The argument in the rebuttal (training is a stochastic process) does not justify treating the model's own predictions as evidence.
> >
> > My reasoning is as follows:
> > (1) Bayesian inference relies on updating a prior belief using observed data (evidence) from the external world.
> > (2) As illustrated by the "biased coin" analogy in my initial review, using a model's own prediction to update its belief is circular. The stochasticity of the model does not transform an internal prediction into external, empirical data.
> > (3) Since the "evidence" is generated internally, the fundamental logic of Bayesian inference does not apply here.
> >
> > This leaves two possibilities:
> > (A) The method is indeed Bayesian. If so, the authors should demonstrate how a model's internal prediction can be validly viewed as independent, external experimental data. The current explanation regarding the stochastic nature of training is insufficient to bridge this gap.
> > (B) The method is not Bayesian. If so, it seems this method uses an update rule that mathematically resembles a Bayesian update but lacks the required epistemological structure. Effectively, it is a heuristic loss function that works well experimentally.
> >
> > If case (B) is true, I recommend the authors to remove the term "Bayesian" from the paper. Using mathematical formalism to dress up a heuristic creates an illusion of theoretical rigor that is misleading to the community.

---

> ### Author Response · Authors · 2025-11-26
> **Clarification on the Bayesian Framework and Evidence Definition**
>
> We thank the reviewer for their continued engagement and for clarifying their earlier question concerning the epistemological basis of our method. In the following, we address the reviewer’s binary formulation directly and illustrate why a Bayesian interpretation is appropriate for our method.
>
> **1. BSD as Recursive Bayesian Estimation**
>
> Although we agree that the standard, static Bayesian view is not fully applicable, we argue that the method isn't simply heuristic and that the Bayesian terminology is not inaccurate. Instead of the static view, BSD can be more accurately described through Recursive Bayesian Estimation (Filtering).
>
> * We treat the network not as a static generator, but as a noisy sensor observing the external dataset $\mathcal{D}$. This perspective is similar to established frameworks in that it views neural network training as Recursive Bayesian Estimation (Singhal & Wu, 1989).
> * At epoch $t$, the network provides a prediction $\hat{y}^t$. This is a noisy measurement of the true state $y$, conditioned on the external input $x$.
> * We use these noisy measurements to iteratively refine the target distribution $y$.
>
> While the sensor (model) evolves during training (making the measurement noise non-stationary), this dynamic is consistent with adaptive filtering and probabilistic self-training frameworks (such as Expectation-Maximization). The update rule is not a heuristic loss but a recursive integration of noisy signals to approximate the posterior. Since SGD dynamics approximate a posterior distribution over model parameters (Mandt et al., 2017), the resulting predictions serve samples from the predictive distribution. This validates treating $\hat{y}^t$ as a noisy measurement of the external input $x$ rather than a fixed internal belief.
>
> **2. The Source of Evidence is External and Independent**
>
> We believe the interpretation of circularity stems from the analogy used. The "imagined coin flip" implies that no information from the physical world enters the system, which would indeed be circular. We propose a different analogy that we believe more accurately reflects our method.
>
> **The Picket Fence Analogy:**
>
> Let's imagine a passenger in a moving car trying to view an object (the true class $z$) obscured by a picket fence.
> * The passenger looks at the real object (the external input image $x$), but their view at any split second is obstructed by the fence (stochastic noise).
> * A single glimpse ($\hat{y}^t$) is noisy and incomplete due to this training-time stochasticity.
> * However, by aggregating these glimpses over time (the Bayesian update), the passenger constructs a clear, stable mental representation of the object.
>
> Importantly, the evidence comes from the object shining through the fence (the external data $x$), not from the observer's imagination. The "Biased Coin" analogy assumes the passenger is closing their eyes and imagining the object, which would indeed be circular.
>
> **3. Capturing dark knowledge via independent evidence accumulation**
>
> To understand why this captures valid evidence, we note that the Dirichlet distribution $Dir(\alpha)$ can be generated by normalizing a set of independent Gamma variables $v_k \sim Gamma(\alpha_k, 1)$ such that $y_k = v_k / \sum v_j$.
>
> * This allows us to interpret BSD as maintaining independent evidence counters (the Gamma variables $v_k$) for each class prior to normalization.
>
> * When an input image $x$ triggers a non-zero prediction for a secondary class (e.g., "cat" features in a "dog" image), this acts as an independent sensor updating the specific underlying Gamma component for "cat".
>
> Assuming that the model learns to extract relevant feature representations during training, the updates capture semantically meaningful information ("dark knowledge") rather than model hallucinations. This assumption is substantiated by our experiments in Section 4.1, which demonstrate that semantically meaningful knowledge emerges during training.
>
> In conclusion, we model the evidence in BSD as external because it depends on the input $x$, as read by the noisy sensor $f$. We clarify that the "evidence" is not the prediction $\hat{y}$ in isolation, but the noisy sensor’s response $f(x)$ to the independent, external input.
>
> To improve the presentation of our method, we have revised Section 3 to explicitly frame BSD as Recursive Bayesian Estimation, clarifying that $\hat{y}^t$ serves as a noisy measurement conditional on the external input $x$, distinguishing our method from circular self-confirmation. We hope that this clarifies that the use of the term “Bayesian” accurately reflects the underlying probabilistic evidence aggregation.
>
> **References**
>
> [1] S., & Wu, L. (1989). Training multilayer perceptrons with the extended Kalman algorithm. Advances in neural information processing systems, 1.
>
> [2] Mandt, S., Hoffman, M. D., & Blei, D. M. (2017). Stochastic gradient descent as approximate Bayesian inference. JMLR, 18(134), 1-35.

---

### Official Review · Reviewer_ht5v · 2025-11-07

**Soundness:** 3
**Presentation:** 3
**Contribution:** 1
**Rating:** 4
**Confidence:** 4

**Summary:**

This paper studies bayesian self-distillation for image classification. It takes a bayesian version of the original targets per image during the training process in order to improve calibration and generalization.

**Strengths:**

* Some of the empirical findings are interesting. For example, Figure 3 that shows that the proposed method results in finding semantic patterns between different classes is interesting. Another interesting finding is Figure 6, where it is shown that the proposed method mitigates the double descent phenomenon. Although this feature is also appearing using another baseline method called Temporal Ensembling (TE) and is not unique to the proposed method.
* Empirically, the proposed method outperforms other baseline, particularly, the closest one in nature which is the progressive self-knowledge distillation (PS-KD). This is shown both in terms of generalization (Table 1), and in my opinion, more so in terms of improvements brought to calibration (Table 2).

**Weaknesses:**

* The proposed method lacks novelty and sufficient reasoning on the main differences with the baseline counterparts. For example, an explanation of the motivation behind switching from PS-KD to the proposed method is missing. Why instead of the teacher model the original targets are used? Why this would be a better training strategy. The current empirical analysis to highlight the main differences between the proposed method and the similar baseline methods is insufficient and not convincing.
* The empirical results are very limited for the scope of the paper. The standard deviation are missing from the tabular results. For example, if the authors observe that “The performance gains are most pronounced on the more complex datasets” it would be worth investigating this further to verify the statement beyond just 3 datasets.

**Questions:**

The main question to the authors, as mentioned in the above section, is the motivation behind the proposed algorithm and a deeper comparison with other similar methods.

---

> ### Author Response · Authors · 2025-11-22
> **We thank Reviewer ht5v for reviewing our work**
>
> We thank the reviewer for their thoughtful feedback and for recognizing the empirical findings regarding semantic patterns and the mitigation of double descent. We appreciate the opportunity to clarify the motivation and expand the empirical scope.
>
> **1. Motivation and Comparison with Baselines (PS-KD)**
> The reviewer asked for a clearer explanation of the motivation behind switching from PS-KD to BSD and a deeper comparison. We have addressed this in the revised version of the paper by explicitly formalizing how existing methods (PS-KD, DLB, TE) can be viewed as special cases of the BSD framework and added a justification for switching from PS-KD.
>
> * **Variance Reduction (lines 45-47):** We added the argument that PS-KD relies on epoch-wise targets, which introduces high variance under augmentations. BSD aggregates predictions over time, smoothing out this variance.
> * **Theoretical Connection (Section 3.3):** We explicitly show how baselines relate to BSD: For instance, PS-KD corresponds to $\gamma=0$ (no memory), and conventional training corresponds to $c \to \infty$ (fixed targets).
>
> **2. Empirical Scope and ImageNet**
> Following the reviewer’s suggestion to investigate performance beyond the initial three datasets, we have conducted additional experiments on ImageNet. The results are included in Table 1 of the revised paper.
>
> * **Results:** BSD achieves $79.47$% top-1 accuracy with ResNet-152, outperforming both the baseline ($78.55$%) and the best performing related method (PS-KD, $78.99$%). This demonstrates that BSD scales effectively to more complex datasets.
>
> **3. Novelty and Justification**
> Regarding the novelty and the question of why we use original targets instead of a teacher:
>
> * **Novelty:** We argue for the novelty of BSD in that it establishes a principled Bayesian framework that treats targets as distributions to be inferred rather than fixed labels. Crucially, it completely decouples from hard targets after initialization, unlike related methods like TE or PS-KD which retain a dependency on one-hot labels.
> * **Efficiency vs. Teachers:** We do not use a teacher model because BSD aims to be an efficient self-distillation method. Training a teacher increases the computational cost. BSD leverages the model's own evolving "dark knowledge" to improve itself in a single training run without the overhead of an external teacher. Additionally, in Table 1 we observe that BSD achieves better results than conventional teacher-student knowledge distillation, likely thanks to the refinement of targets over epochs.
>
> **4. Standard Deviations**
> Regarding the standard deviations, we would like to clarify that we omitted them in the tables solely to maintain readability within the strict space limits. Following the reviewer’s suggestion, we have included the standard deviations in the tables in the revised paper, but we would like to emphasize that our conclusions already considered this variability.
>
> We thank the reviewer again for their time and insights. We hope that the revised paper and this response have adequately addressed the reviewer's concerns.

---

> > ### Author Response · Authors · 2025-11-27
> > **Follow-up on baseline differentiation and ImageNet experiments**
> >
> > We thank the reviewer once again for their helpful comments. We would like to ask the reviewer to get back to us if their concerns have not been fully addressed. Specifically, we wanted to highlight that we have revised the method description to clarify how baselines relate to BSD (Section 3.3) and have expanded empirical results to include ImageNet experiments (Table 1) in the revised paper to address the reviewer’s concerns regarding motivation and empirical scope. We remain available to answer any further questions or provide additional clarifications before the discussion period closes.

---

### Author Response · Authors · 2025-12-03
**Author Final Remarks**

We thank the AC for their time and the reviewers for their feedback. We are pleased that the reviewers found our method to be novel (aqga, s5Y5), principled (aqga, s5Y5, cnVX), theoretically elegant (s5Y5), and efficient (cnVX, s5Y5), while yielding improvements with respect to calibration, generalization, and robustness (ht5v, aqga, s5Y5) compared to existing self-distillation methods. Our rebuttal focused on clarifying the conceptual positioning of the method and validating the generalizability of our claims across larger scales and more applications.

## Main contributions
* We propose Bayesian Self-Distillation (BSD), a principled method that constructs sample-specific target distributions via Bayesian inference, allowing the training process to operate independently of hard targets after initialization. This avoids ad-hoc modifications to the training procedure, ensures the method remains theoretically grounded and promotes interpretability.
* We provide the first theoretical formalization of dark knowledge (Hinton, 2014), i.e. the nuanced inter-class and sample-specific information contained in the predictions of a neural network, and demonstrate that BSD accentuates its emergence.
* Extensive experiments show that BSD consistently outperforms existing self-distillation methods in generalization, calibration and robustness.

## Summary of changes, additions and clarifications
### Empirical Validation
* **Scalability to ImageNet (ht5v, s5Y5, cnVX):** To validate the method's scalability, we have provided results on ImageNet in **Table 1**. BSD achieves **79.47%** top-1 accuracy (ResNet-152), outperforming both the baseline (78.55%) and the best performing related method (PS-KD, 78.99%), confirming that BSD scales to larger datasets.
* **OOD Detection and Calibration (cnVX):** We extended the evaluation to include Out-of-Distribution (OOD) detection (**Table 6**) and calibration under covariate shift (**Figure 15**). In both settings, BSD demonstrates superior results compared to baselines.

### Theoretical Clarifications
* **Bayesian Justification (aqga):** We have revised **Section 3.2** to provide additional intuition while clarifying the epistemological basis of the method by explicitly framing BSD as Recursive Bayesian Estimation (filtering), addressing potential concerns regarding circularity. Reviewer aqga stated they would 'raise [their] score' if this point was addressed. We trust that our clarification, including the Picket Fence Analogy, satisfies the condition.
* **Relation to Self-Distillation Baselines (ht5v):** We have formalized the connection between BSD and existing methods (PS-KD, TE, DLB) in **Section 3.4**, demonstrating how previous approaches can be viewed as special cases of our framework, and clarified that a limitation of PS-KD is the variance induced by relying on a single epoch to provide the target (**lines 45-47**).
* **The choice of loss function (cnVX):** While reviewer cnVX proposed that we experiment with calibration-specific objectives, we have clarified that minimizing KL-divergence naturally aligns with matching the aggregated target distributions, and that BSD achieves lower ECE and higher accuracy than any other calibration methods tested (**Table 7**).
* **Error amplification (s5Y5):** We have clarified that we do not observe error amplification and expanded the discussion in **Section 4.2.5** to include that the poor selection of hyperparameters may cause underfitting.

### Clarifications on Experiments and Results
* **Comparisons with Temporal Ensembling (s5Y5):** We have highlighted that while Temporal Ensembling (TE) is comparable at low levels of label noise, BSD offers superior robustness at high levels (**Table 4**).
* **Comparisons with Label Smoothing and Practical Utility (cnVX):** We would like to draw the AC's attention to the fact that Reviewer cnVX may have overlooked Table 7, present in the original submission. The concerns regarding Label Smoothing/Mixup and practical gains are addressed explicitly in **Table 7**, where BSD outperforms not only these methods but the SOTA by reducing ECE by over 13%, SCE by over 5%, and ACE by 20%.
* **Standard Deviations and Code (ht5v, s5Y5):** We have added standard deviations to all tables and released the source code.

We believe our clarifications and experimental validation fully address the reviewers' concerns, and reinforce the paper's original contributions. Specifically, we have validated our method on the large-scale ImageNet dataset (ht5v, s5Y5, cnVX) and in the OOD setting (cnVX) as requested. Additionally, we have clarified that the baseline comparisons (cnVX) were present in the original submission, where BSD displays significant improvements, and have provided additional intuition regarding the conceptual positioning of the method (aqga, ht5v). We view these changes as minor modifications that overall improve clarity of presentation. We wish to thank all reviewers for their constructive comments.

---

> ### Author Response · Authors · 2025-12-03
> **References**
>
> [1] Hinton, G., Vinyals, O., and Dean, J. "Dark knowledge." Presented as the keynote in BayLearn, 2014.
>
> [2] Hinton, G., Vinyals, O., and Dean, J. "Distilling the Knowledge in a Neural Network." Deep Learning and Representation Learning Workshop in Conjunction with NIPS. 2014.

---

### Meta-Review · Area_Chair_SJVa · 2025-12-08

**Summary:**

This paper proposes Bayesian Self-Distillation (BSD), a self-distillation method that constructs evolving soft targets using Dirichlet updates derived from the model's own predictions. BSD aims to eliminate dependence on hard labels after initialization and to leverage dark knowledge through Bayesian evidence accumulation. The method is theoretically motivated by recursive Bayesian estimation and empirically evaluated on CIFAR-10/100 and Tiny ImageNet.

Most reviewers agree that the idea is interesting. However, multiple concerns were consistently raised. For example, there is no sufficient conceptual justification for treating model predictions as Bayesian evidence, raised by reviewer aqga. The novelty relative to existing self-distillation baselines is limited, raised by reviewer ht5v. The improvements relative to simpler alternatives, such as label smoothing or Mixup, are limited, raised by reviewer cnVX. Large-scale and OOD evaluation are missing, raised by reviewers ht5v, s5Y5, and cnVX. Concerns about potential error amplification, raised by reviewer s5Y5.

The rebuttal provided by the authors successfully addressed partial empirical-scope concerns by adding ImageNet, OOD evaluation, and standard deviations. Also, the error-amplification behavior has been clarified. Reviewers acknowledged these additions. However, the most critical concern raised by reviewer aqga, i.e., foundational Bayesian justification, remains.  Also, reviewer cnVX mentioned that more comprehensive experimental validation is required in the current version.

With most reviews below the threshold and remaining doubt about both soundness and practical gains, the authors are encouraged to clarify and address these issues in a new cycle.

**Reviewer Concerns:**

Concerns addressed by the rebuttal:

1. Empirical scope and scalability (ht5v, s5Y5, cnVX): ImageNet results were added.
2. Missing comparisons to calibration baselines (cnVX): Add comparisons in Table 7.
3. Standard deviations and code availability (ht5v, s5Y5): Now included.
4. Error-amplification concerns (s5Y5): Authors expanded analysis.

Concerns not adequately addressed by the rebuttal:

1. Novelty vs. PS-KD (ht5v).
2. Foundational Bayesian justification (aqga).
3. Limited improvement and insufficient validation (cnVX).

**Reviewer Scores:**

Based on the rebuttal and discussion, the reviewers are unlikely to increase their scores.

Reviewer ht5v: 4 (unchanged) The reviewer’s empirical concerns were addressed. However, concerns about novelty seem only partially alleviated.

Reviewer aqga: 4 (unchanged) The reviewer explicitly states that "my primary concern regarding the soundness of the Bayesian framework remains unaddressed" after rebuttal.

Reviewer s5Y5: 6 (unchanged) Concerns about large-scale validation and error amplification could be addressed. The reviewer seemed generally positive.

Reviewer cnVX: 4 (unchanged) After rebuttal, the reviewer states clearly: "I prefer to keep my original score." Concerns about practical gain and completeness of experiments remain.

---

### Decision · Program_Chairs · 2026-01-26

Reject